# Sensitivities of Simulated Mixed-phase Arctic Multilayer Clouds to Primary and Secondary Ice Processes

Gabriella Wallentin[1], Annika Oertel[1], Luisa Ickes[2], Peggy Achtert[3], Matthias Tesche[3], and Corinna Hoose[1]

[1]Institute for Meteorology and Climate Research Troposphere Research (IMKTRO), Karlsruhe Institute of Technology (KIT), Karlsruhe, Germany
[2]Department of Space, Earth and Environment, Chalmers University, Gothenburg, Sweden
[3]Leipzig Institute for Meteorology, Leipzig University, Leipzig, Germany

**Correspondence:** Gabriella Wallentin (gabriella.wallentin@kit.edu)

**Abstract.** Multilayered clouds are frequent in the Arctic but their detailed analysis is underrepresented. Here, we simulate two cases observed during the 2019/2020 MOSAiC expedition using the ICosahedral Non-hydrostatic (ICON) model to explore the most accurate representation of these multilayer clouds. With a limited area setup, we investigate how cloud layers respond to perturbations in cloud droplet activation, primary ice, and secondary ice production (SIP). Using the measured aerosol concentration, we constrain our model through a new immersion freezing parameterisation. We find that multilayered clouds are challenging to simulate in remote areas with sparsely assimilated thermodynamics and that large-scale biases in the global forcing carry over to high-resolution simulations. In terms of cloud microphysics, high-temperature ice-nucleating particles (INPs) are necessary to capture the cloud phase of warm mixed-phase clouds. However, constraining the model to the observed INPs is insufficient; a factor of $10^6$ is required to reach observed ice mass concentrations, which is also achieved by including SIP. Breakup upon ice-ice collisions is explosive and can increase the cloud ice number concentration by a factor of $10^6$. Furthermore, the seeder-feeder mechanism significantly boosts snowfall by a factor of $10^3$. An accurate representation of these microphysical processes is crucial to simulate multilayer clouds.

## 1 Introduction

Mixed-phase clouds are ubiquitous in the Arctic climate system (Shupe et al., 2005). While a large observational foundation has been built over the last two decades (e.g. Shupe and Matrosov, 2006; Shupe, 2011; Morrison et al., 2012) their representation in climate models (Tan and Storelvmo, 2019), regional cloud-resolving models (Klein et al., 2009; Morrison et al., 2009; Fu et al., 2019; Schäfer et al., 2024), and large eddy simulation (LES) models (Ovchinnikov et al., 2014; Bulatovic et al., 2023; Kiszler et al., 2023) remains challenging.

Phase partitioning in Arctic single-layer mixed-phase clouds is commonly observed as a liquid cloud layer precipitating ice (Solomon et al., 2018). The cloud phase is governed by temperature and the competing growth mechanisms of liquid droplets and ice crystals. Clouds are commonly found to be purely liquid above 0°C, where growth mechanisms lead to rain formation. Meanwhile, super-cooled liquid water and ice can coexist in clouds within the mixed-phase temperature regime between -38°C

and 0°C. Due to the lower saturation vapour pressure over ice, frozen hydrometeors such as cloud ice, snow, graupel, and hail may grow by vapour deposition at the expense of evaporating liquid droplets in an environment that is sub-saturated with respect to relative humidity (RH) over water but saturated with respect to RH over ice. Commonly known as the Wegener-Bergeron-Findeisen process (WBF) (Wegener, 1911; Bergeron, 1928; Findeisen, 1938; Korolev, 2007) this process may lead to the glaciation of the cloud under suitable environmental conditions with intermediate vertical velocities.

In the mixed-phase cloud regime, the relative abundance of cloud ice and liquid droplets is highly dependent on the presence of aerosols. Cloud condensation nuclei (CCN) act as catalysts for droplet formation by reducing the supersaturation required for condensation of water vapour. These CCN are hydrophilic and/or soluble aerosols. In the Arctic, species such as locally emitted sea salt and sulfate from dimethyl sulfate (DMS) (Schmale et al., 2021) as well as anthropogenic sulfates (Udisti et al., 2016) and aged black carbon (Zieger et al., 2023) from long-range transport are the main contributors to cloud droplet formation. In general, the concentration of these species in the Arctic is very low, sometimes even forcing the dissipation of clouds in this aerosol-limited environment (Mauritsen et al., 2011; Bulatovic et al., 2023; Sterzinger et al., 2022).

Ice nucleating particles (INP) are aerosols that are typically insoluble and are the reason ice is observed at temperatures above the freezing temperature of pure water droplets (at approximately $-38$°C (Mossop, 1954; Korolev et al., 2017)). Acting as a seed for ice nucleation, water vapour may deposit onto the INP to form ice through the deposition nucleation mechanism, while INPs immersed within liquid droplets may freeze through immersion freezing (Hoose and Möhler, 2012). The INPs in the Arctic region are commonly mineral dust transported from the south, and biological (heat labile) or organic aerosols (Creamean et al., 2022) emitted with sea spray (DeMott et al., 2016). Previously, the CCN and INP concentrations in the Arctic have only been known from shorter expeditions. The Multidisciplinary drifting Observatory for the Study of Arctic Climate (MOSAiC) expedition during 2019/2020 (Shupe et al., 2022) now for the first time provides one-year-long surface-based concentrations of CCN (Koontz et al., 2019; Dada et al., 2022; Bergner et al., 2023) and INP (Creamean et al., 2022).

Secondary ice production (SIP) includes processes whereby multiple ice fragments are generated through interactions between frozen hydrometeors or with liquid droplets. SIP has been studied in laboratory experiments (Korolev and Leisner, 2020) but its atmospheric relevance remains unclear. Out of the at least six hypothesised mechanisms, three have been identified as possible major contributors based on modelling studies: (i) rime-splintering (also known as the Hallet-Mossop process) (Hallet and Mossop S.C, 1974), (ii) collisional breakup upon ice-ice collisions (Takahashi et al., 1995), and (iii) droplet freezing and shattering (Mason and Maybank, 1960). Rime-splintering, parameterised after Hallet and Mossop (1974), describes the production of small ice fragments during the adhesion and subsequent freezing of super-cooled liquid droplets when colliding with frozen hydrometeors (riming). The Hallet-Mossop process is implemented in several global models (Komurcu et al., 2014) and many cloud-resolving models (e.g. Morrison et al., 2005; Seifert and Beheng, 2006). The droplet shattering mechanism describes the ejection of ice crystals from large supercooled cloud droplets during phase change as a result of pressure build-up (Mason and Maybank, 1960; Kleinheins et al., 2021). The collisional breakup mechanism describes the fracturing of frozen hydrometeors upon collisions with other ice particles. These three mechanisms have been shown to occur in Arctic clouds (Pasquier et al., 2022). They are often considered the main reason for model discrepancies in ice number concentrations in Arctic clouds (Sotiropoulou et al., 2020; Zhao et al., 2021; Possner et al., 2024).

Multilayer clouds (MLCs) are defined here as vertically stacked cloud layers that are separated by a sub-saturated layer with respect to ice. MLCs have a global occurrence frequency of about 20%-30% (Subrahmanyam and Kumar, 2017; Wang et al., 2016; Liu et al., 2012; L'Ecuyer et al., 2019) but have been found to be more common in the Arctic region (Herman and Goody, 1976; Intrieri et al., 2002; Vassel et al., 2019; Nomokonova et al., 2019; Vüllers et al., 2021). For instance, data from the MOSAiC campaign show a 51% occurrence of liquid-bearing MLCs (Silber and Shupe, 2022). Arctic MLCs have been studied in idealised setups (e.g. Herman and Goody, 1976; Harrington et al., 1999; Luo et al., 2008; Chen et al., 2020; Bulatovic et al., 2023) and in more detail through a model intercomparison case study from the Mixed-Phase Arctic Cloud Experiment (M-PACE) campaign near the north slope of Alaska (Morrison et al., 2009). However, this intercomparison study explored individual liquid layers within the same cloud, rather than the separated cloud layers we focus on here. Earlier studies have shown that MLCs may form through the incomplete dissipation of a stratus cloud (Herman and Goody, 1976), due to large-scale advection (Luo et al., 2008; Morrison et al., 2009), or through the moistening and cooling of the atmosphere caused by ice precipitation (melting and sublimation) in regions of weak supersaturation (Harrington et al., 1999). The cloud layers within the MLC system interact through radiation, when the presence of an upper layer reduces the radiative cooling of the lower layer (Shupe et al., 2013; Christensen et al., 2013; Turner et al., 2018; Lonardi et al., 2022). They also interact through microphysical processes such as the seeder-feeder mechanism in which frozen hydrometeors sediment from an upper layer into a lower cloud layer. This can lead to efficient dissipation of the lower cloud through the WBF mechanism (Dedekind et al., 2024; Proske et al., 2021). The impact can be further enhanced by SIP (Georgakaki et al., 2022). This natural seeder-feeder mechanism may thus increase precipitation (Jian et al., 2022; Dedekind et al., 2024). For MLCs in the Arctic, the frequency and atmospheric conditions of this process are still largely unknown.

This study aims to build on the existing knowledge of Arctic multilayer clouds and to further investigate these cloud systems in a detailed and realistic perspective using the ICosahedral Non-hydrostatic (ICON) model in a limited-area mode. We investigate the limits to accurately model the layering of the clouds in a remote region, evaluated with the observations from the MOSAiC campaign. We perturb the aerosol parameterisations in an effort to 1) obtain the observed state, and 2) understand sensitivities in the cloud response to aerosols. For this, we use the observed aerosol concentrations to constrain the CCN and INP parameterisations to better represent the Arctic aerosols. SIP is explored and the impacts of seeding and glaciation due to the WBF mechanism are discussed. The paper is structured as follows. The observational data from the MOSAiC campaign, used for model constraint, and evaluation, is introduced in Sect. 2. The implemented constraints are then introduced in the model setup, Sect. 3, together with the microphysical parameterisations. Section 4 describes the case study and the synoptic situation. The result sections include a general evaluation of the model in Sect. 5.1 and a more detailed high-resolution evaluation of the model together with the sensitivity studies performed, in Sect. 5.2 and Sect. 5.3. The results from SIP simulations are shown in Sect. 5.4. Finally, a discussion is provided in Sect. 6.

## 2 The MOSAiC campaign and observational data

The observational data, used for model comparison and aerosol constraint, is collected from the MOSAiC campaign (Shupe et al., 2022) where the ice breaker *RV Polarstern* (Knust, 2017) was moored to an ice floe in the high Arctic during 2019/2020. For a complete list of instruments available during the campaign please refer to Shupe et al. (2022). Atmospheric variables are taken from level 3 files produced from Vaisala RS41 6-hourly radiosondes (Maturilli et al., 2022). Cloud variables are collected from the Cloudnet database (Engelmann et al., 2023) retrieved from ground-based remote sensing instruments through the approach by Illingworth (2007). Liquid water content (LWC) is calculated where liquid water path (LWP) is retrieved by a Microwave Radiometer (MWR) and the Cloudnet classification algorithm flags liquid water. The uncertainty in the LWC is 15-25 % (Frisch, 1998; Griesche et al., 2024). The ice water content (IWC) is calculated from the 35 GHz radar reflectivity factor and temperature from a forecast model using the approach by Hogan et al. (2006) with an associated uncertainty of + 40 % and - 30 %. Cloud classification is a challenging topic. During the days investigated here, both radar and lidar products were available for Cloudnet retrievals. In general, the highest classification confidence is given for clouds that are detected by both lidar and radar. Confidence decreases for the upper clouds in the MLC systems considered here as the lidar signal is attenuated by the lowermost cloud. For identifying MLCs, however, we utilise soundings (see Sect. 4) and confirm the presence of clouds in saturated layers through lidar observations.

CCN and INP concentrations are used to scale the model representation of cloud droplet activation and heterogeneous freezing (Sect. 3.2). Aerosol data are obtained from measurements from the Swiss aerosol container (Beck et al., 2022) and the US Atmospheric Radiation Measurements (ARM) facility (Creamean et al., 2021a; Koontz et al., 2019). Surface INP concentrations during MOSAiC were measured using total aerosol filter samples, collected every 72 hours and analysed offline using the Colorado State University (CSU) Ice Spectrometer (IS; Cremean et al.(2022). The INP concentration during the case study is shown in Fig. 1 together with the parameterisation for immersion freezing in the model by Hande et al. (2015). Surface-based measurements of CCN concentrations (Koontz et al., 2019) were perturbed by local ship emissions (due to the sampling close to the ship plume) and were thus filtered using a pollution mask from Beck et al. (2022). The mean surface CCN concentration (for the chosen case study, as described in Sect. 4) at $1\,\%$ supersaturation is shown in Fig. 2 (marked with $\ast$), together with the original and modified parameterisations (see Sect. 3.2).

## 3 Model setup

The ICosahedral Non-hydrostatic (ICON) model (Zängl et al., 2015) is the operational forecast model of the Deutscher Wetterdienst (DWD), Germany's National Meteorological Service, since 2015 and is developed jointly by DWD, KIT, Max-Planck Institute for Meteorology (MPI-M), Deutsche Klimarechenzentrum (DKRZ), and Center for Climate Systems Modeling (C2SM). It runs on an icosahedral grid structure; a triangular grid structure projected onto the globe, efficiently removing the pole singularity problem that models with spherical coordinates struggle with (Purser, 1988). ICON has been used for global as well as regional simulations in the Arctic with multiple nested domains (e.g. Kretzschmar et al., 2020; Kiszler et al., 2023). Microphysical perturbation studies on Arctic clouds have also been evaluated using the ICON model and its predecessor Consortium

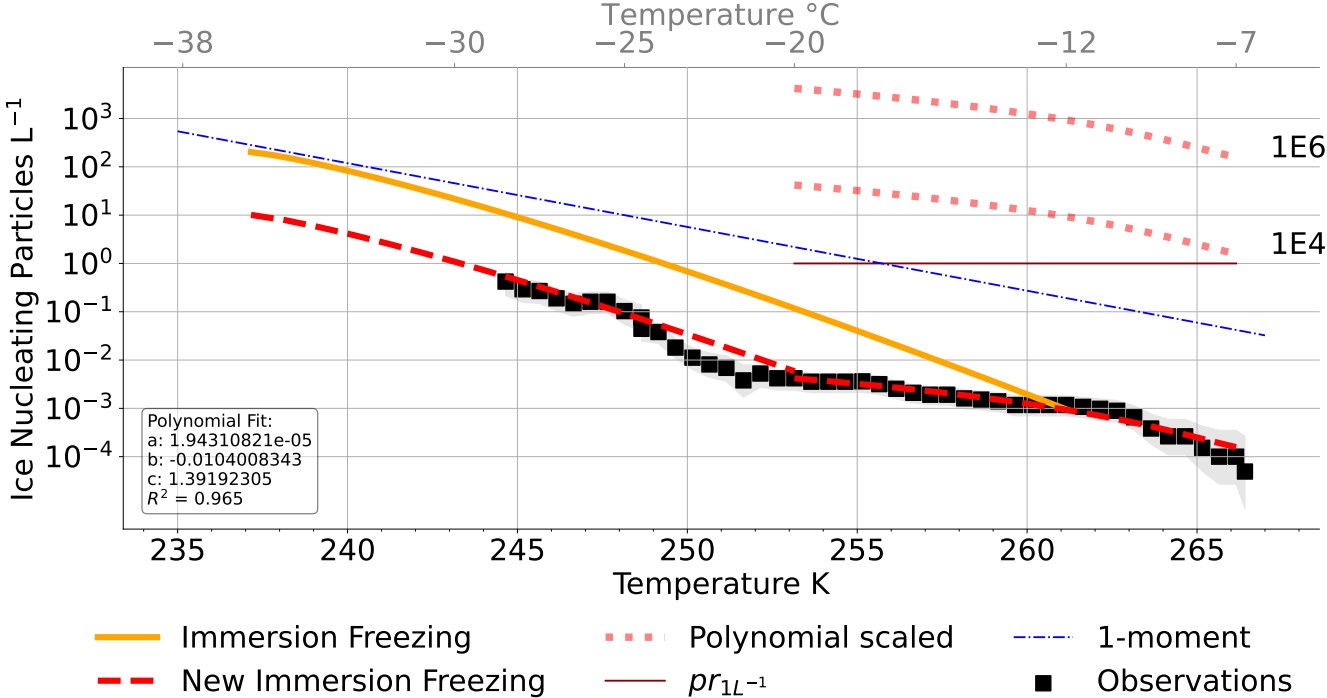

**Figure 1.** Ice nucleating particle concentration ($L^{-1}$) plotted against temperature. The immersion freezing parameterisation by Hande et al. (2015) is marked with solid orange and the observational data set (Creamean et al., 2022) in black squares with its confidence intervals indicated in shading. The observationally constrained parameterisation is shown (dashed red) where at temperatures below -20°C the original immersion parameterisation is scaled down by 0.05 while at high temperatures -20°C < T < -7°C a polynomial fit to the observational data is used with parameters indicated in the box. Scalings to the warm INPs are shown (small dotted red) marked with 1E4 and 1E6. $pr_{1L^{-1}}$ signifies one of the simulations performed with a constant prescribed INP concentration at 1 $L^{-1}$ at temperatures -20°C < T < -7°C. Finally, the 1-moment cloud ice parameterisation by Cooper (1986) is shown in dashed blue.

for Small-scale Modelling (COSMO), which included the same cloud microphysics scheme (e.g. Stevens et al., 2018; Possner et al., 2017; Loewe et al., 2017; Possner et al., 2024). Here we make use of the triangular grid to create domains encompassing 90°N. To study clouds, we set up a case study, making use of ICON Global 13 km analysis as initial and boundary conditions.

Atmospheric variables as well as mass concentration of hydrometeors are introduced at the domain boundaries at intervals of three hours. We use ICON version 2.6.5 with a semi-implicit time integration solver. The model is run in an offline nested limited area mode with nests at 6 km (ICON domain R3B8) and a cloud-resolving scale of 1.6 km (R2B10). Further nested simulations were also run at 400 m (R2B12) and 100 m (R2B14) horizontal grid spacings, marked out in Fig. 3. Each nest is provided with an initial state and boundary conditions from the output of the coarser simulation.

At a grid spacing of 6 km, convection is parameterised (Tiedtke, 1989; Bechtold et al., 2008) while at higher resolutions both deep and shallow convection are considered explicitly resolved. The model top is kept at 23 km while the sleeve coordinates

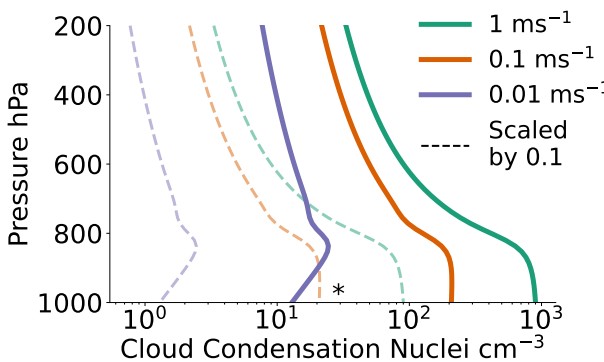

**Figure 2.** CCN parameterisation by Hande et al. (2016) (solid lines) giving CCN concentration ($\mathrm{cm}^{-3}$) for each pressure level parameterised with respect to the vertical velocity. The observational mean value at 1 % supersaturation (Koontz et al., 2019) is marked with a $^*$ and the scaled parameterisation (divided by 10) in dashed lines.

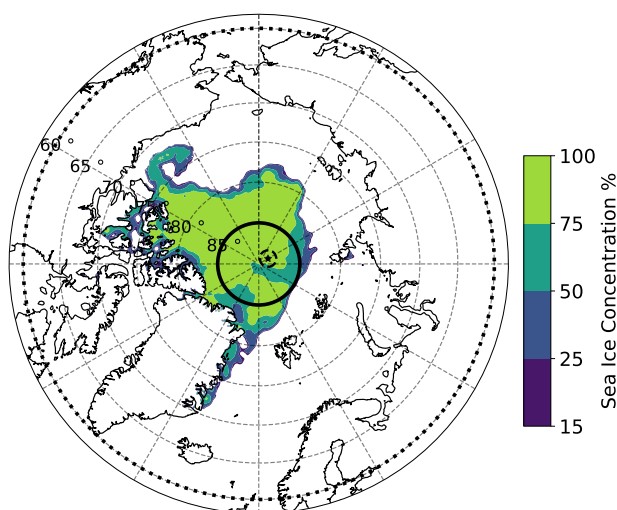

**Figure 3.** 1st of September sea ice concentration with the nested domains; dotted 6 km domain extending from $60°$N -$90°$N. A solid line marks the 1.6 km nest spanning $85°$N - $90°$N and a dashed marks the 400 m nest. The 100 m nest would be covered by the average daily location of the ship marked with a black star and is thus not shown. The 400 m and 100 m nests are adjusted to centre on the average daily position of the ship. The 6 km domain spans $60°$N-$90°$N and the 1.6 km domain $85°$N-$90°$N. The 400 m domain has a radius of 112 km and is centred on the ship location. The 100 m domains also follow the ship location with a 33 km radius.

space the model levels to keep the highest density of levels in the lowest kilometres. The 6 km and 1.6 km simulations are kept to 90 levels while the 400 m (100 m) simulation is increased to 150 (200) vertical levels. This translates into a vertical grid spacing at the lower cloud top ($\sim 600$ m) of about 55 m for the 6 km and the 1.6 km simulations. The 400 m and 100 m simulations have 39 m, and 32 m vertical grid spacing, respectively, at a similar height. At 100 m horizontal grid spacing,


turbulence is partially resolved using the Lilly-Smagorinsky parameterisation (Smagorinsky, 1963; Lilly, 1962). In contrast, turbulent diffusion at coarser grid spacing is by default represented by a $2^{nd}$ order closure scheme (Raschendorfer, 2001). The time step for physics (including microphysics, turbulent diffusion, and saturation adjustments) is set to 50 s, 12 s, 4 s and 0.8 s, respectively. The radiation is described by the ecRad radiation scheme (Hogan and Bozzo, 2018). We use the two-moment
microphysics from Seifert & Beheng (2006) with six hydrometeor classes, namely cloud droplets, rain, graupel, hail, snow, and cloud ice. The two moments refer to the first moment of the particle size distribution, the mass concentration, and the zeroth moment, the number concentration. The use of a two-moment scheme can in general improve the representation of the phase partitioning in mixed-phased clouds (Seifert and Beheng, 2006). Hourly accumulated microphysical process rates are collected with the model output and used as diagnostic indicators for which processes contribute to layer growth as well as cloud phase
partitioning. The model is initialised at 00 UTC for the 6-km nest, and the other nests are initialised at 04 UTC, 06 UTC, and 08 UTC, respectively, to account for model spin-up at a coarser resolution before further increasing the horizontal and vertical resolution. This relatively short spin-up time was deemed necessary to not substantially deviate from the observed thermodynamic state. The 400 m nests (and subsequent 100 m) are initialised by the 1.6-km output, specifically the prescribed $1\,\mathrm{L}^{-1}$ simulation (see Sect. 3.2). The simulations on a finer resolution are shown in Appendix A. Further increases in resolution
beyond 1.6 km have large impacts on the cloud droplet number concentration (Fig. A2a) due to the better resolved vertical velocities. Thus, we achieve higher activation rates with finer grid spacing. The cloud ice is less affected than the cloud liquid in this specific case, but an increase is seen with the 100 m simulation. The high computational demand of running simulations at a fine resolution means that improvements, in particular concerning cloud ice, are insufficient to justify further study of microphysical perturbations with this configuration. It is hence concluded that the 1.6-km grid spacing is a good compromise
between resolution and computational costs.

## 3.1 Treatment of cloud microphysical processes

### 3.1.1 Cloud droplet formation

The parameterisation by Hande et al. (2016), developed for ICON using its predecessor, COSMO, is used for the cloud condensation nuclei (CCN) activation. This CCN activation scheme is based on the High Definition Clouds and Precipitation for
advancing Climate Prediction (HD(CP)$^2$) Observational Prototype Experiment (HOPE) in 2013 and builds on a previous CCN parameterisation by Abdul-Razzak and Ghan (2000) and simulated aerosol concentrations. CCN activation is parameterised with respect to vertical velocity and is thus highly resolution-dependent. Perturbations to study the MLC sensitivity to CCN are explored in this paper and the parameterisation is shown in Fig. 2 together with the scaling towards observations.

### 3.1.2 Primary cloud ice formation

There are five major pathways of primary ice production in ICON, three heterogeneous and two homogeneous nucleation modes. The three heterogeneous ice nucleation pathways are deposition nucleation, immersion freezing of cloud droplets, and the freezing of rain droplets. Deposition nucleation is the instant freezing of vapour deposited onto an aerosol while immersion

freezing is the freezing of a cloud droplet containing an insoluble INP. Deposition nucleation and immersion freezing are parameterised based on simulated aerosols for Germany (Hande et al., 2015), both require ice supersaturation and activate at temperatures between -53°C < T < -20°C and -36°C < T < -12°C, respectively. Immersion freezing takes precedence wherever there are cloud droplets available. Rain freezing, which also includes the freezing of drizzle drops, is counted as a source of primary ice as rain droplets implicitly contain many INPs. The freezing of a raindrop is described by Bigg (1953) and is based on probabilities of freezing depending on the volume of the droplet and temperature. Rain freeze is the only ice production mechanism above -12°C currently implemented in the model, Sect. 3.2 introduces the changes performed on the immersion freezing parameterisation to improve on this representation. Contact freezing is not considered.

### 3.1.3 Secondary ice production

Secondary ice production includes microphysical processes where fragments of ice are generated from existing ice. SIP effectively increases ice number concentrations and is hypothesised to "fill the gap" between observed INPs and measured ice crystal number concentrations as well as between measured and modelled ice crystal number concentrations (Sotiropoulou et al., 2020; Zhao and Liu, 2022). Three SIP pathways have been implemented in the configuration of ICON used for this study: droplet freezing and shattering, rime-splintering (Hallet-Mossop process), and breakup upon ice-ice collisions (Han et al., 2024). The clouds investigated here exist in the temperature range of rime-splintering (-8°C < T < -3°C), collisional breakup (-35°C < T < 0°C), and droplet shattering (T < 0°C). Thus, all three mechanisms are evaluated. Rime-splintering is already included in the reference setup for ICON. This process has been found to be weak when acting alone but in combination with ice multiplication from breakup upon ice-ice collisions, it has been shown to have a considerable impact in simulations of Arctic clouds (Sotiropoulou et al., 2020; Schäfer et al., 2024). Rime-splintering is parameterised based on the observations of Hallet and Mossop (1974). The droplet shattering mechanism, describing the process where large supercooled cloud droplets may fragment into smaller ice particles during freezing, is parameterised by Sullivan et al. (2018). This process is parameterised within the primary ice production parameterisation for rain (and drizzle) freeze and will inherently depend on the rate of rain freeze. The collisional breakup mechanism requires collisions in the cloud whereby impacts between ice and snow, graupel or hail, plus all the combinations of these (except ice-ice collisions), produce ice fragments. The breakup collision mechanism is investigated using the parameterisation developed for COSMO by Sullivan et al. (2018) based on the laboratory work of Takahashi et al. (1995). This parameterisation has been questioned for its atmospheric efficiency (Sotiropoulou et al., 2021; Dedekind et al., 2021; Georgakaki et al., 2022; Han et al., 2024) due to the original experiment being conducted with large (1.8 cm diameter) colliding particles. To this effect we further investigate a breakup collision parameterisation based on Takahashi et al. (1995) with a diameter scaling used in multiple studies (e.g., Sotiropoulou et al., 2021; Georgakaki et al., 2022; Han et al., 2024). For detailed implementation in ICON please refer to Han et al. (2024).

### 3.2 Constraining the microphysical parameterisations

The phase partitioning of a cloud is highly dependent on the microphysical parameterisations. We hypothesise that the INPs must be adequately represented so as not to overestimate the impact of heterogeneous ice nucleation on ice clouds at low

temperatures. Similarly, the INP species active at higher temperatures need to be included to achieve a mixed-phase state at warmer, but sub-zero, temperatures. In the 6 km simulation, using a 1-moment microphysics scheme, the ice crystal number concentration is simply parameterised by an exponential function depending on temperature (Fig. 1) (Cooper, 1986). The output from this nest is used to simulate the high-resolution simulation where we can make use of the 2-moment scheme in ICON. In this setup, both mass and number concentrations are traced in time and we can use more sophisticated parameterisations of primary ice production to better represent mixed-phase clouds (Seifert and Beheng, 2006).

Due to the lack of an INP parameterisation developed for the Arctic, we use the ground-based observations from the MO-SAiC campaign (see Sect. 2) to constrain the primary ice production. We constrain the three heterogeneous freezing parameterisations; deposition nucleation, immersion freezing and rain freeze. These parameterisations act at different temperatures as specified in Sect. 3.1.2. At cold temperatures, (T < -20°C), all three are active and are thus scaled similarly by a factor of 0.05, following the comparison to the observational values of INPs (see Fig. 1) at these temperatures. At warmer temperatures, (T > -20°C) only immersion freezing and rain freeze may be active. To better represent warm mixed-phase clouds we adjust the immersion freezing parameterisation and add INPs at high temperatures (-20°C < T < -7°C), fitting a second-order polynomial to the observed INP. Rain freeze is, however, kept scaled down for all temperatures to limit the impact of the increased rain production for scaled CCN concentrations. INPs activating as immersion nuclei at temperatures above (below) -20°C will from now on be called warm (cold) INPs. To summarise, the parameterisations for deposition nucleation and rain freeze are scaled down throughout their active temperature range while immersion freezing is treated differently for cold (scaled down) and warm (fitted polynomial) temperatures.

To investigate whether these constraints are appropriate, sensitivity studies have been performed and are tabulated in Table 1. These are applied to the nest with a grid spacing of 1.6 km, initialised and updated through the boundaries with the output from the 6 km simulation. To obtain the observed state and explore the sensitivity of the clouds to changes in aerosols, perturbations to the parameterisations are performed.

Sensitivity tests on the primary ice production include the constrained parameterisation introduced above named "INP" in Table 1 and further scalings to the polynomial fit in the immersion freezing parameterisation at warm temperatures, these are indicated by "warm" in the table. A factor of 1E4 and 1E6 is applied to the polynomial, increasing the immersion freezing at "warm" temperatures between -20°C < T < -7°C. Furthermore, a prescribed INP concentration of 1 $L^{-1}$ ($pr_{1L^{-1}}$ simulation) has also been applied for a uniform immersion nuclei concentration at warm temperatures. Meanwhile, at cold temperatures the heterogeneous processes are scaled down (by $10^{-3}$, effectively to zero), this setup is further studied with a CCN scaling to explore the sensitivity to cloud droplet activation ($pr_{1L^{-1}}$ +CCN).

Secondary ice processes are also explored through various SIP simulations. The five SIP simulations tabulated include droplet shattering and collisional breakup (with all simulations including rime-splintering). For the breakup upon collision mechanism, we apply the parameterisation by Takahashi et al. (1995) and further explore the diameter scaling introduced in Sect. 5.4. The first SIP simulation is performed with the original primary ice parameterisations (1.6km + SIP). Three other SIP setups are performed using the new INP parameterisation for primary ice production (SIP, SIP scaled, INP+CCN+SIP scaled). This includes the SIP addition on the constrained INP simulation (SIP simulation) and a similar setup with a diameter

**Table 1.** Sensitivity simulations on the 1st and 3rd of September 2020 at 1.6 km grid spacing. BR and DS signify the SIP breakup upon collision and droplet shattering respectively while rime splintering is included in all simulations. "Warm" refers to the temperature range $-20°C < T < -7°C$. For more details please refer to the text.

| Name | INP | CCN | SIP | Day |
|---|---|---|---|---|
| 1.6km | original | | | 1st & 3rd |
| INP | constrained | | | 1st & 3rd |
| $pr_{1L^{-1}}$ | 1 L$^{-1}$ at warm T | | | 1st & 3rd |
| $pr_{1L^{-1}}$ + CCN | 1 L$^{-1}$ at warm T | 0.1 | | 3rd |
| INPx1E4 | constrained + warm**x**1E4 | | | 3rd |
| INPx1E6 | constrained + warm**x**1E6 | | | 3rd |
| 1.6km + SIP | original | | BR & DS | 3rd |
| SIP | constrained | | BR & DS | 3rd |
| SIP scaled | constrained | | BR scaled & DS | 3rd |
| $pr_{1L^{-1}}$ + SIP | 1 L$^{-1}$ at warm T | | BR & DS | 3rd |
| INP+CCN+SIP scaled | constrained | 0.1 | BR scaled & DS | 3rd |

scaling within the breakup routine (SIP scaled). We further explore a CCN scaling to this setup (constrained INP and scaled breakup upon ice-ice collisions) to explore the impact of large supercooled droplets on the droplet shattering mechanism (INP+CCN+SIP scaled). To be emphasised here, the droplet shattering mechanism is tuned down due to the tuning of the rain freeze mechanism, where the droplet shattering is implemented. To investigate the primary ice impact on SIP we further use a prescribed INP concentration of 1 L$^{-1}$ (at $-20°C < T < -7°C$) and apply the Takahashi et al. (1995) breakup parameterisation ($pr_{1L^{-1}}$ +SIP).

## 4   Case description

The 1st to the 3rd of September 2020 (from now on called the 1st, 2nd, and 3rd), during the MOSAiC campaign, were chosen to investigate two different multilayer cloud systems. On the 1st, a high ice cloud overlaying a mixed-phase boundary layer cloud can be seen in Fig. 4a showing the derived LWC and IWC products from Cloudnet (Engelmann et al., 2023). The 2nd comes with a shallow lower layer and a shorter mixed-phase cloud overlapping. On the 3rd, a double-layered system can be seen in the lowest kilometre of the atmosphere with some sporadic third layers (around 2-4 km). Crosses (circles), in Fig. 4a, mark the cloud base (top) for each layer identified using the MLC algorithm by Vassel et al. (2019). This algorithm uses radiosonde data and radar input to determine if a profile contains multiple layers and can distinguish cloud layers remarkably well. Discrepancies may be due to drifting radiosondes and precipitation (falling ice crystals), which complicates the determination of cloud boundaries. We define MLCs as cloud layers with an interstitial layer subsaturated with respect to ice with a separation of at least 100 m, following Tjernström and Graversen's (2009) determination of the lower tropospheric thermal structure. No

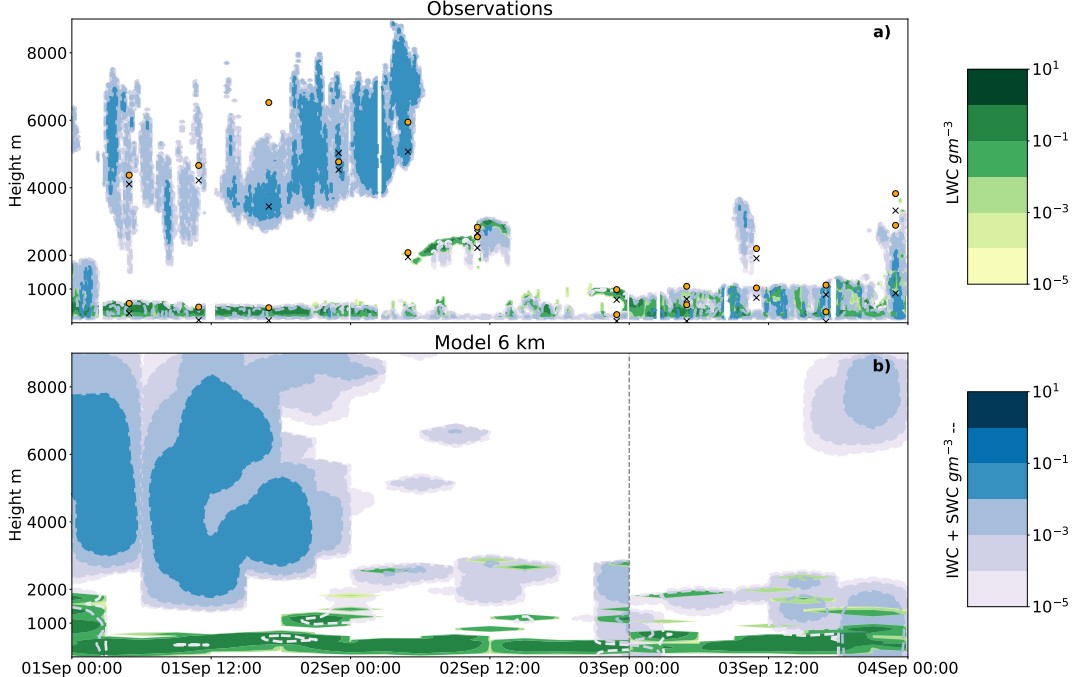

**Figure 4.** 3-day case study in September 2020, with the liquid water mass content (green) and cloud ice water mass content (filled contours in blue and dashed outlines). (a) 10 min observational mean, (b) 3-hourly model output at 6 km grid spacing, here also the snow water content from the model is included. The vertical line partitions the two simulations performed. Crosses (circles) in (a) mark the cloud base (top) for each layer identified using the MLC algorithm by Vassel et al. (2019). The missing cloud top at 00 UTC on the 2nd exceeds the plotted region.

phase distinction is applied in the classification (all overlapping clouds are accounted for independent of phase). For detailed descriptions of the observational algorithm please refer to the original paper by Vassel et al. (2019).

## 4.1 Synoptic situation

The synoptic situation from the model output at 00 UTC on the 1st can be seen in Fig. 5a. Temperatures are usual for the time of the year with the sea ice temperature locked to 273 K due to the melting of the sea ice and mean sea level pressure is shown in contours. To understand air mass origin, 10-h backward trajectories are calculated with LAGRANTO (Wernli and Davies, 1997; Sprenger and Wernli, 2015) based on hourly 3D wind fields from the 6 km simulation. Trajectories are initialised hourly between 10 UTC and 12 UTC near the ship location. Back trajectories initialised at 12 UTC on the 1st (Fig. 5b) show upper-level air parcels moving faster towards the ship compared to the < 4 km trajectories, which spend more time over the pack-ice. Interestingly, on the 3rd, air parcel trajectories below 2 km height move faster and originate from further away than trajectories at higher altitudes, which is likely related to the presence of low-level jets (López-García et al., 2022). Low-level

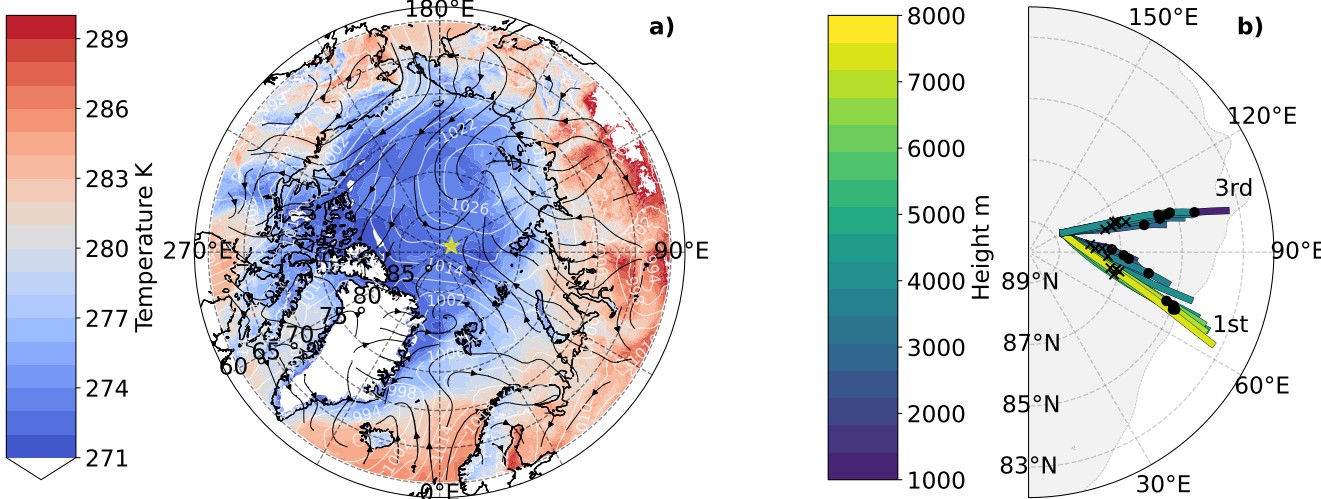

**Figure 5.** (a) Modelled synoptic situation on the 1st September. Mean sea level pressure contours are marked in white with wind streamlines (black arrows) showing the relative magnitude and direction of the wind at 10 m. The filled contours show surface temperature. The average daily location of the ship is shown with a star (*) (1st; 88.3°N, 117°E, 3rd; 88.7°N, 120°E). (b) Back trajectories initialised at the daily average ship location at 12 UTC for the 1st and 3rd of September, the dates are marked at the end of the trajectory cluster. Trajectories are initialised hourly between 10 UTC and 12 UTC near the ship location every 6 km within 114°E-120°E and 88.3°N-88.5°N on the 1st. On the third, the trajectories are initialised within 118°E-120°E and 88.5°N-89°N. Vertical trajectory starting positions are set to every 250 m in the lowest 2 km, every 500 m between 2 to 7 km height, and finally every 1 km up to 10 km height. Colours indicate the height at which the back trajectory is initialised and crosses (dots) mark 8 UTC (4 UTC). For the 3rd, only trajectories initialised at 4 km or lower are shown because of the low clouds simulated. Additionally, the extent of the sea ice concentration (>15%) is shown with a grey contour.

winds, with magnitudes in the lowest km of up to 15 ms$^{-1}$, primarily flow towards the ship location from the Siberian coast, shifting from approximately parallel to the 70°E meridian on the 1st to 100°E meridian on the 3rd.

## 5 Results

### 5.1 Modelling MLCs

A first reference simulation is performed at 6 km grid spacing to evaluate the model against observations. Although a side-by-side comparison is difficult due to differences in spatial and temporal output, general features can be assessed. Figure 4 shows a time-height cross-section from an Eulerian perspective. The water content derived from the observations (Fig. 4a) is compared to the model output (Fig. 4b). The model runs from 00 UTC on the 1st of September 2020 until 00 UTC on the 3rd. Another simulation is then initialised at 00 UTC on the 3rd to prevent the model from deviating too far from the global analysis. The first 6 hours are considered spin-up and are not included in mean state analyses but are shown here for initialisation

comparison. The mean state is calculated for grid points within a radius of 15 km of the ship to account for the daily drift of the ship and spatial variability. Overall, the location and timing of cloud formation in the simulation agree well with the observations. The boundary layer cloud on the 1st is well represented, although the phase is predominantly liquid, unlike the observed mixed-phase character. The large vertical extent of the upper cloud on the 1st may be the result of sparse vertical levels at altitudes above 7 km together with a lack of precipitation formation (few sinks). Moisture profiles compare well with the observations (Fig. B1a-b). During the 2nd the cloud layers are less persistent in the model compared to the observations but the layers are captured at adequate heights. Due to the brief overlap of the cloud layers, this day will not be further studied in this paper. On the 3rd, two layers of clouds are simulated, one at around 600 m and the second at approximately 2 km. The lower layer cloud is well constrained to the boundary layer but liquid dominated while the upper cloud is placed 1 km higher than the observations. In general, the 6 km simulation accurately represents the clouds but falls short in representing the phase partitioning. Constraints have been implemented in the microphysical parameterisations to address this issue (see next section).

The placement of these layers depends on the initialisation of the model. DWD assimilated the radiosondes from MOSAiC including the temperature, wind profiles, and relative humidity. However, data assimilation is a difficult field and capturing a perfect vertical thermodynamic profile is challenging. On the 1st (Fig 6a), the model quite accurately captures three low-temperature inversions (at 300 m, 1000 m, and 1800 m) and correctly predicts the loss of the upper two in favour of a persistent layer at 600 m in the 12 UTC radiosonde profile (Fig. 6b). The 3rd (Fig. 6c) is initialised with a smoother profile than observed and inversions are less accurately captured. The temperature profile at 12 UTC on the 3rd (Fig. 6d) deviates quite radically from the observed radiosonde profile. ICON is found to place inversions too high in comparison with observations, a similar error has been found to be due to excessive vertical mixing in the European Centre for Medium-Range Weather Forecasts (ECMWF) Integrated Forecast System (IFS) model (Sandu et al., 2013). Moisture content and relative humidity with respect to liquid water are shown in Fig. B1 for completeness. Subsequent high-resolution simulations (see Appendix A) do not improve on the main layering (some spurious intermittent layers may occur with higher vertical resolution), as this is mostly governed by the initial state.

## 5.2   1.6 km simulations for 1st of September

The 1st of September 2020 is further simulated with a higher resolution nest at 1.6 km, where the boundary conditions (and initialisation) are provided with input from the 6 km simulation. We make use of the 2-moment microphysics and aerosol constraints laid out in Sect. 3.2. The high-resolution simulation, similarly to the 6 km simulation, produces a thick ice cloud and a fully liquid lower layer, found at a temperature just below 0°C. A vertical cross-section of hydrometeors with time is shown in Fig. 7 for the original setup at 1.6 km (Fig. 7b) and the simulation with constrained INP (Fig. 7c). Compared to the observations (Fig. 7a), the model in its original setup produces too much ice in the upper layer while the lower layer is dominantly liquid compared to the observational mixed-phase character.

With the INP constraint at cold temperatures (including the reduction of immersion freezing, deposition nucleation and rain freeze) and additions to the immersion freezing routine for warm INPs, surprisingly, no large impacts can be seen (Fig. 7c). The upper layer shows small changes due to the scalings of the heterogeneous parameterisations. A reduction in INPs (at cold

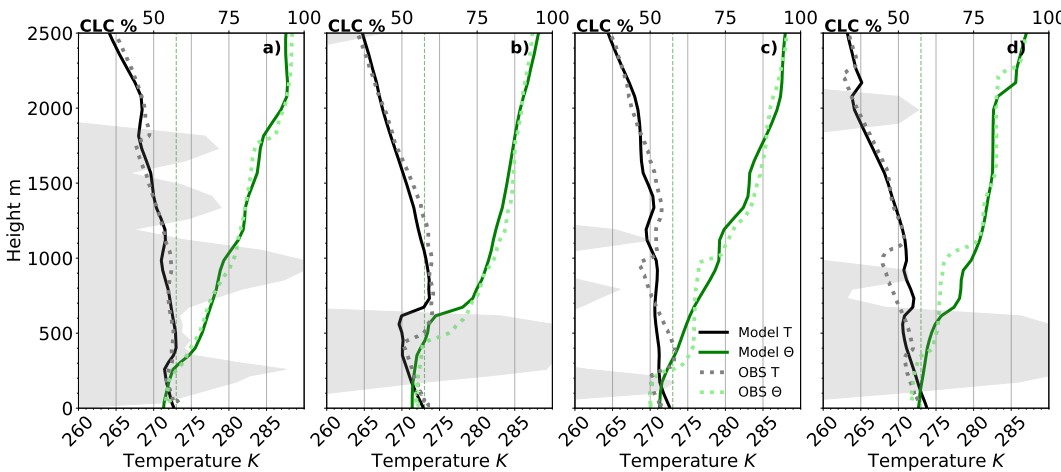

**Figure 6.** Vertical profiles of the lowest layer(s) for temperature (black), potential temperature ($\theta$) (green), and the modelled cloud cover (CLC) (contours in temperature plot) signifying the cloud layers. (a) 1st at initialisation time 00 UTC, (b) 1st at 12 UTC, (c) 3rd at initialisation time 00 UTC, and (d) 3rd at 12 UTC. The observational data (in dots) are from the radiosondes (Maturilli et al., 2022) and the model data (solid lines) are the 6 km grid spacing simulation, averaged over two grid cells within the 15 km radius of the daily average ship position. The dotted vertical line marks 273 K.

temperatures above 4 km, see Fig. 8e) induces small reductions in ice mass concentration in the upper levels of the upper cloud (above 4 km). This amounts to a time-averaged (10 UTC-12 UTC) reduction by a factor of 0.5 in the ice mass concentration (Fig. A4c). In the lower levels of the upper cloud (at warmer temperatures), an increase is noted. The time-averaged difference in the ice mass concentration amounts to an increase by a factor of 1.4. The ice number concentration for the INP simulation across the upper layer is decreased by a factor of 8 and reaches a maximum value of 0.4 $L^{-1}$ (Fig. A4d). The lower layer is unaffected by the changes due to its relatively warm temperature (Fig. 6a,b). Prescribing INPs at 1 $L^{-1}$, ($pr_{1L^{-1}}$, Fig. 7d), effectively increasing the immersion freezing at the bottom of the upper layer while reducing INPs at colder temperatures, renders similar cloud features with the largest difference being the increase in precipitation during 10 UTC-12 UTC. Figure 8 shows microphysical process rates for the sensitivity studies. An increase in ice mass concentration below 4000 m, in the $pr_{1L^{-1}}$ simulation (Fig. A4c), shows the impact of increased vapour deposition (Fig. 8c) onto cloud ice rendering larger particles capable of efficient aggregation into snow as seen in Fig. 8f. Ice-ice collisions result in particles assigned to the snow category, while ice-snow collisions aggregate into larger snow particles (Fig. 8f). The snow produced in the lower part of the upper cloud seed the lower cloud (Fig. 7d) but not efficiently enough to glaciate or strengthen it. The snow particles seemingly fall through the lower layer with negligible interactions. Due to its warm cloud top temperature, the lower layer may only become mixed-phase with the presence of seeding, however, glaciation is also inefficient at these high temperatures. We do not observe local seeding in the observations and we must assume that previous seeding of the lower layer has occurred or colder temperatures that allowed for primary ice nucleation upstream. Uncertainties in the cloud ice retrieval may also be causing this

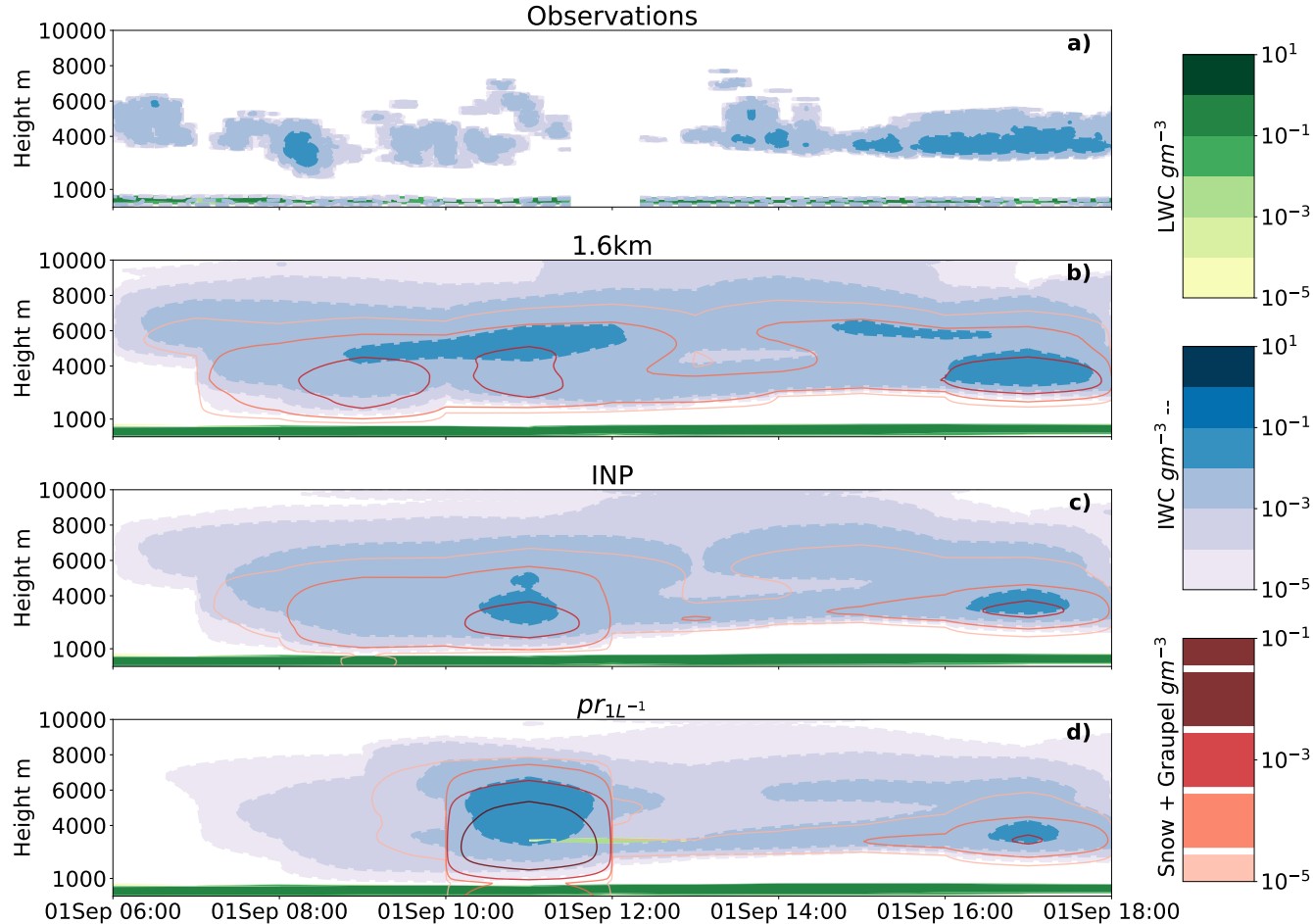

**Figure 7.** Vertical cross-section on 1st September 2020, with the liquid water mass content (green) and cloud ice water mass content (filled contours in blue and dashed outlines) in (a) 10 min observational mean together with (b) reference simulation, hourly model output at 1.6 km grid spacing, (c) the constrained INP simulation, and d) the prescribed $1\,\mathrm{L}^{-1}$ INP at warm temperatures. Additional model parameters shown are the graupel and snow mass content in red (solid lines).

discrepancy. Due to its high temperature, further studies on primary and secondary ice processes in the lower cloud layer are thus deemed unnecessary.

## 5.3 1.6 km simulations for 3rd of September

The 3rd of September 2020 is also simulated at a finer grid spacing of 1.6 km using the 2-moment microphysics. The time-height contours of the 3rd are shown in Fig. 9, where the reference at 1.6 km (Fig. 9b) and the constrained INP simulation (Fig. 9c) are shown together with the observations.

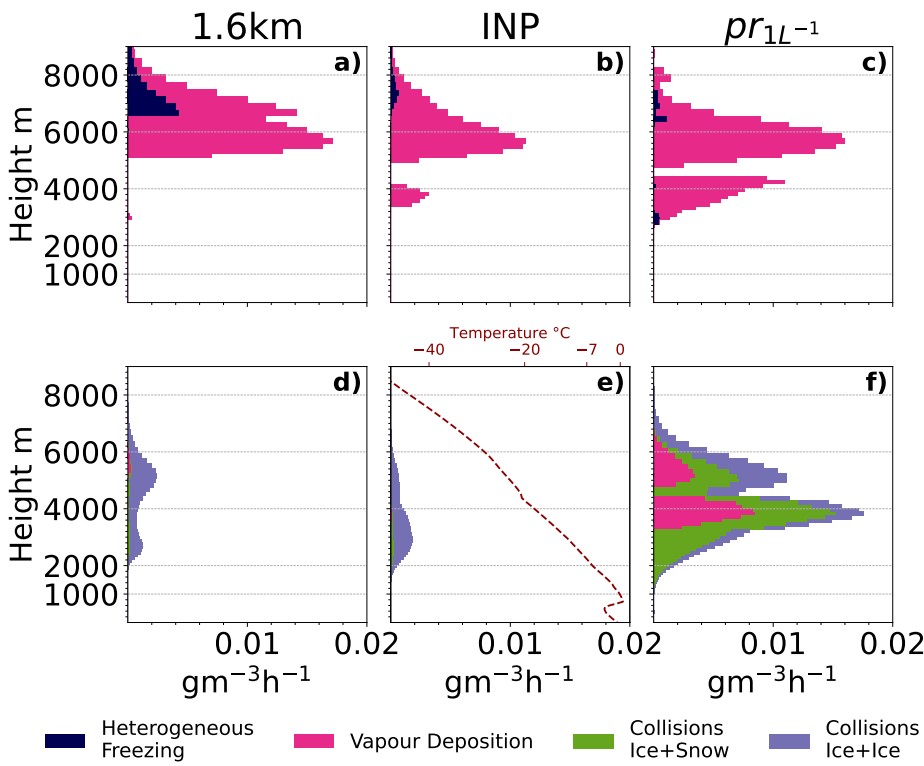

**Figure 8.** Microphysical process rates hourly accumulated during the simulation and here taken as a mean between 10 UTC-12 UTC on the 1st of September. a-c) show processes for cloud ice, and d-f) for snow. Colours correspond to the various process rates defined in the legend. Collisions lead to aggregation where ice+ice and ice+snow give aggregates of snow. Additionally, a temperature profile is provided in (e) to more easily distinguish between temperature regions of cold (T < -20°C) and warm (-20°C < T < -7°C) INP scalings within the parameterisations for primary ice production.

As previously discussed, the model layers the clouds at the wrong heights, but more striking is the lack of ice in the model. At a mean cloud-top temperature of -8°C (Fig. 6d), the only primary ice parameterisation active in the reference is rain freeze, however, the addition of warm INPs (Fig. 9c) does little to increase the ice mass concentration. Blue contours marking the cloud ice mass concentration can be seen to only slightly increase (mostly just after 06 UTC) when including warm INPs. Time-averaged (06 UTC-12 UTC) maximum ice mass concentration increases by a factor of 1.5 (for vertical profiles see Fig. A2d). This small impact does not reach the observed levels of ice. Thus, we explore the scalings of the primary ice parameterisations in the model. Multiplying the warm INP polynomial (Fig. 1) by 1E4 (Fig. A1a) and 1E6 (Fig. 9d), give rise to more similar values to the observations. These are very extreme scalings, but as the observed INP concentration is low, this gives INP concentrations between 1-100 L$^{-1}$, on the upper end in midlatitudes and very high for Arctic clouds (Porter et al., 2022; Raif et al., 2024). With this high INP concentration, the ice number concentration increases as well. Time-averaged (06 UTC-12 UTC) maximum ice number concentrations in the upper cloud layer reach 0.22 L$^{-1}$ and 6.7 L$^{-1}$ for the INPx1E4

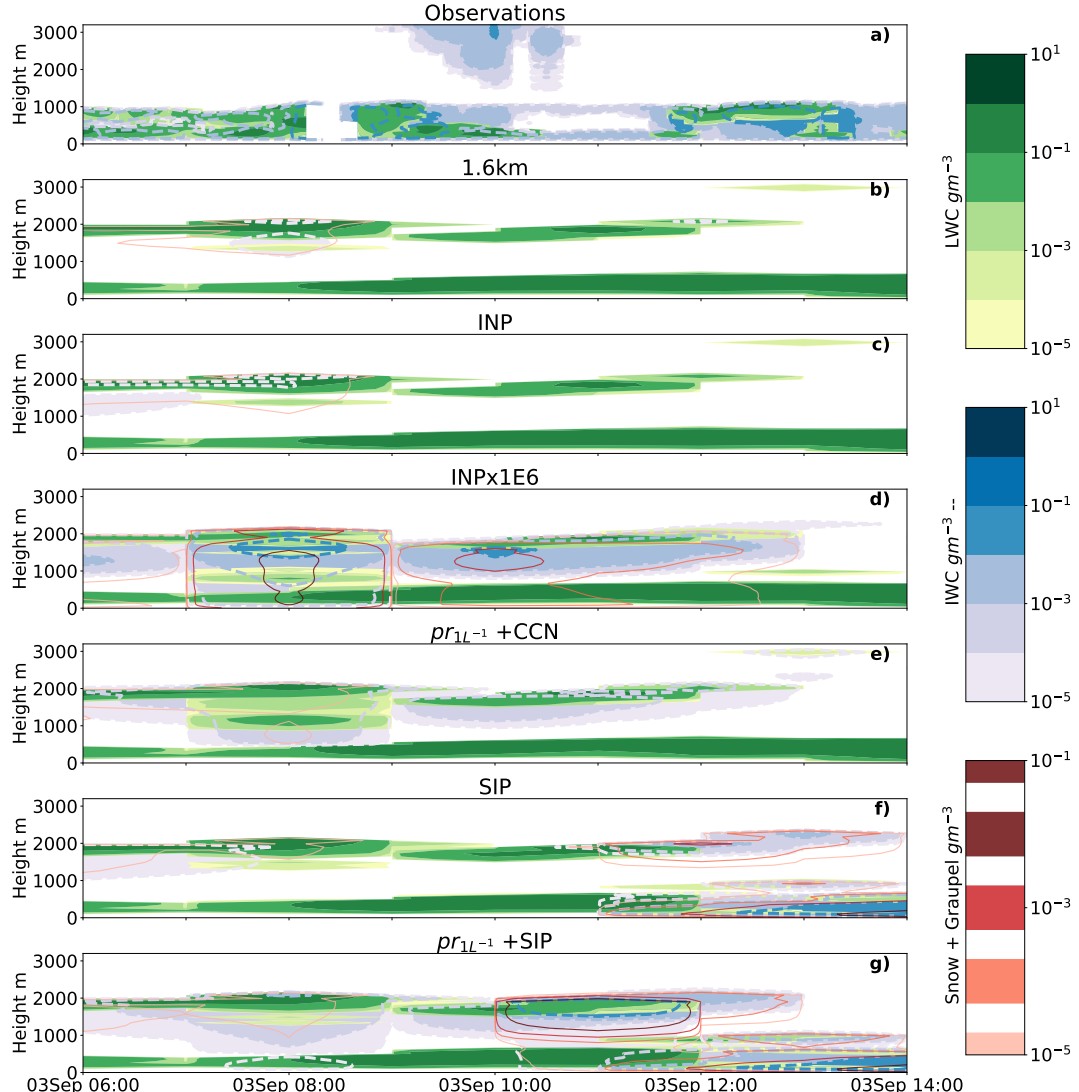

**Figure 9.** Time height plot of hydrometeors from the 3rd of September with liquid water mass content (green) and cloud ice water mass content (filled contours in blue and dashed outlines). (a) Observations 10 min mean, (b) reference simulation, (c) INP constrained simulation, (d) warm INP scaling by 1E6, e) a prescribed INP concentration of $1\,\mathrm{L}^{-1}$ at warm temperatures combined with a CCN scaling of 0.1, f) SIP using the same primary ice nucleation as the INP simulation with added droplet freezing and shattering and breakup upon ice-ice collisions, g) prescribed INP at $1\,\mathrm{L}^{-1}$ with added droplet freezing and shattering and breakup upon ice-ice collisions. Additional model parameters are the snow and graupel mass content in red (solid line). For clarification on the scalings please refer to the text and Fig. 1.

and INPx1E6 simulations respectively (Fig. 11e). This is compared to $0.0008\,\mathrm{L}^{-1}$ for the INP simulation (Fig. 11e), an increase of more than two and almost four orders of magnitude respectively. However, irrespective of this substantial increase in ice

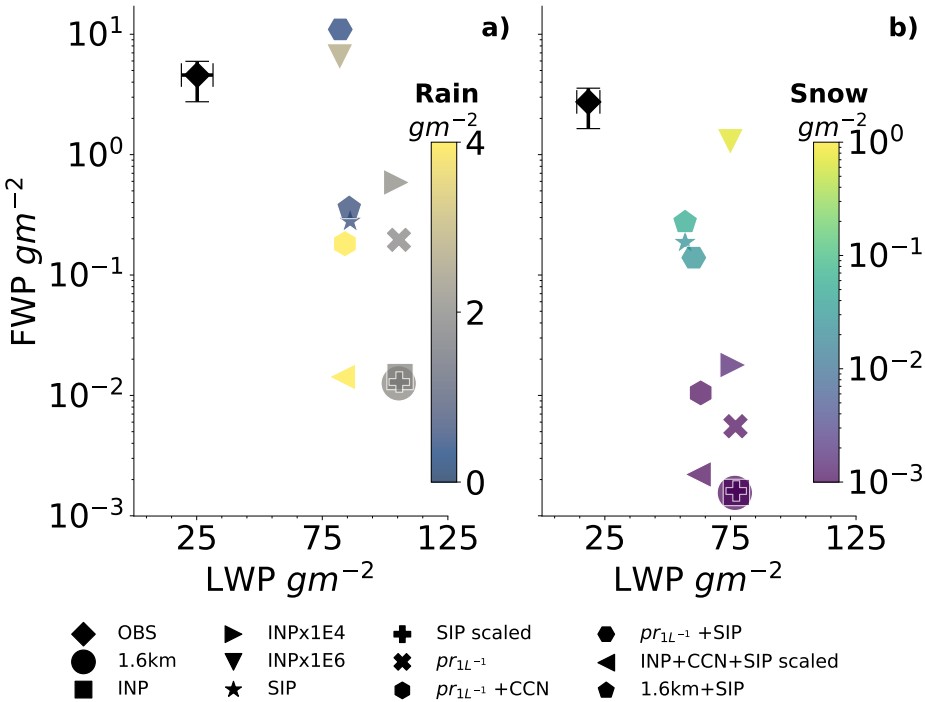

**Figure 10.** Frozen hydrometeor water path (FWP) the vertically integrated sum of cloud ice, snow, graupel and hail, over liquid water path (LWP) for all simulations on the 1.6 km resolution on the 3rd of September between 06 UTC-12 UTC. (a) The column integrated FWP vs LWP with colours indicating the precipitation contribution from the rain in $gm^{-2}$. The observations are integrated up to 1.1 km to exclude the upper ice cloud during this time. (b) FWP and LWP are integrated over the modelled lower layer with an imposed upper threshold at 750 m. The colour bar in (b) shows the snow falling from the lower layer (graupel is negligible).

number concentration, these values do not necessarily lead to the glaciation of the cloud, as seen here and in previous studies
(Stevens et al., 2018; Solomon et al., 2018).

As a scaling of 1E4 to 1E6 is needed to create appreciable ice content in the clouds we further explore a prescribed INP concentration at warm temperatures of 1 per L ($pr_{1L^{-1}}$). The prescribed INP compare well with the 1E4 scaling, with slightly lower values for cloud ice ( Fig. A1c). To investigate the impacts of the cloud droplet activation on the too-thick modelled liquid layer, a sensitivity experiment with a decreased cloud droplet activation rate ($pr_{1L^{-1}}$+ CCN) is performed (Fig. 9e).
The cloud structure changes during 07 UTC -09 UTC where the presence of a third layer is evident, similar to the INPx1E6 simulation. To better compare with the observed values we look at the column integrated values where we include graupel and snow together with ice in a frozen water path (FWP) category. Figure 10 shows the time-averaged (06 UTC-12 UTC) FWP vs LWP.

The large underestimation (more than two orders of magnitude) of FWP in the 1.6 km simulation (circle in Fig. 10) compared
to observed values (diamond) is quite evident while the LWP is approximately four times too high in the model. The small

change using the new INP parameterisation (INP simulation, square) gives an increase in FWP with a preference for cloud ice. The prescribed INP simulation ($pr_{1L^{-1}}$, tilted cross) looks similar in the contours to the 1E4 scaling (right-pointing triangle) but gives a three times lower FWP. This corresponds nicely to the almost three times lower INP concentrations for the prescribed INP simulation at minimum cloud-top temperatures (about -9°C).

As expected with a lower CCN concentration ($pr_{1L^{-1}}$ +CCN, tilted hexagon), we see an increase in rain (colour bar of Fig. 10a). With a lower activation rate of CCN, the cloud droplets may grow to larger sizes due to less competition for vapour and convert more efficiently to rain through autoconversion. With the presence of more cloud liquid, due to the presence of a third cloud layer (Fig. 9e), condensation increases due to the saturation adjustment within this layer (Fig. A2e) and the resulting impact of the CCN scaling is small. When reducing the CCN by a factor of 10, the LWP only drops by a factor of 1.28.

In terms of droplet number concentration, time-averaged (06 UTC-12 UTC) maximum values reach 40 $\mathrm{cm}^{-3}$ for the $pr_{1L^{-1}}$ simulation while a reduction by a factor of three is noted for the $pr_{1L^{-1}}$ +CCN simulation (Fig. A2a).

The large scalings of warm INPs (INPx1E4 and INPx1E6) thoroughly increase the FWP, however highly non-linearly with a 40 (450) times increase between INP and INPx1E4 (INPx1E6). Here, we notice that the lack of glaciation in Fig. 9d may not show the full story. When we look at the integrated values a 77% decrease in LWP in the upper layer between the 1.6 km and

the INPx1E6 (down-pointing triangle) simulation can be seen, showing the WBF process in action together with an increase in riming by cloud droplets (Fig. B2a-b). An increase in riming may be due to the sheer number of ice particles now present in the cloud.

Looking at the lower layer only, Fig. 10b, similar trends to the full integration can be seen with a next-to-none impact of simply adding the warm INPs (INP simulation). The scaling of warm INPs (simulations INPx1E4 and INPx1E6) both increase

the FWP in the lower layer. As this layer is found at a temperature just below 0°C, this increase is due to seeding from above. An increase in precipitation from the lower layer can also be seen with the highest INP scaling (INPx1E6), where snow precipitation increases by three orders of magnitude, showing the seeder-feeder mechanism in play. Interestingly, the 1E4 scaling does not show a similar pattern of precipitation increase. Instead, this simulation shows only a large increase in ice mass concentration. With a larger amount of ice crystals in the upper cloud layer, they largely occupy the same space and

increasing the number of ice crystals in the same volume makes collisions more likely (the collision process rates, where ice collides with ice or snow to form snow are shown in Fig. B2i-j). Snow, with a larger size, tends to fall more readily and allows for seeding of the lower layer, where riming is efficient (Fig. B2j), explaining the larger amount of snow (and thus FWP) for INPx1E6 in Fig. 10b.

## 5.4    Can the lack of cloud ice be explained by SIP?

Secondary ice processes may be the missing link between the observed INP concentrations and the frozen water path, which is underestimated by the simulations discussed so far. We investigate whether the required warm INP scaling of 1E6 detailed above could be replaced by a (potentially) more realistic pathway. On top of the rime-splintering already implemented in the reference, we add breakup upon ice-ice collisions as well as droplet freezing and shattering and investigate the impact of different INP concentrations available for primary ice production. With a mean cloud top temperature of -1.5°C (between

06 UTC -12 UTC) for the lower layer on the 3rd, two out of three SIP (collisional breakup and droplet shattering) may be active, while the upper layer at a mean cloud top temperature of -8.5°C, overlaps with the temperature range for all three (including rime-splintering).

Five simulations have been performed and are tabulated in Table 1. Three simulations explore the primary ice impact on the SIP where the INP concentration is changed from the original (1.6km+SIP) to the constrained primary ice production (SIP

simulation), and finally, set to 1 per L ($pr_{1L^{-1}}$+ SIP). The final two simulations explore a scaled breakup parameterisation, using Takahashi et al. (1995) scaled by the colliding particle diameters (Han et al., 2024). A CCN scaling is performed on top of this to explore its impact on the droplet shattering mechanism.

We start by exploring the impact of primary ice production on SIP. The SIP simulation (Fig. 9f) and the 1.6km+SIP simulation display comparable cloud structures and column-integrated values (Fig. A1e, Fig. 10). The $pr_{1L^{-1}}$+ SIP shows similar

features but with more pronounced ice mass concentration throughout the simulation. All SIP implementations almost fully glaciate the lower layer just after 12 UTC, some liquid persists while the upper layer dissipates similarly to the other experiments (Fig. 9a-e). Figure 11 shows the time-averaged (06 UTC-12 UTC) fragment generation rates from secondary ice production (Fig. 11a-d). The full impact of SIP on cloud ice number concentration is shown in Fig. 11a while Fig. 11b-d show the respective impacts on the cloud ice number concentration from rime splintering, droplet shattering, and breakup upon ice-

ice collisions respectively. The ice number concentration is shown in Fig. 11e and the ice enhancement factor (IEF, see below) is shown in Fig. 11f. The maximum ice number concentrations, Fig. 11e, in the lower cloud layer reach 484 L$^{-1}$, 396 L$^{-1}$, and 156 L$^{-1}$ for the 1.6km+SIP, SIP, and $pr_{1L^{-1}}$+ SIP simulations respectively.

The large increase in ice number concentration drives the increase in vapour deposition through the WBF process (Fig. B2c,f,h) resulting in the glaciation of the lower layer. We find the breakup upon ice-ice collisions to be the dominant mechanism for this

increase in ice number (Fig. 11d) and mass concentrations (Fig. 10). The major contributions are from small ice crystals colliding with snow or snow-snow collisions (not shown). The ice-snow collisions are the most active in-cloud while snow-snow collisions dominate throughout the sub-cloud layer. The rime splintering and droplet shattering show only small contributions (Fig. 11b-c). The breakup upon ice-ice collision parameterisation is highly sensitive to temperature fluctuations and the sudden onset of glaciation through ice particles generated by SIP is due to a drop in temperature of the lower layer (not shown).

One of the main differences between the SIP and $pr_{1L^{-1}}$+SIP simulations is the seeding seen with a higher baseline of INPs. The seeding into the lower layer, seen in Fig. 9g, has a surprisingly small impact on the cloud layer during the time window 06 UTC-12 UTC, referring back to Fig. 10b. The simulations with SIP implementations do not largely differ in LWP nor FWP in the lower layer, showing the lack of interaction with the falling hydrometeors from the upper layer. The initiation of seeding is explained by the higher collision rates as previously discussed for a larger amount of small ice crystals (Fig. B2k,n). The

mean hourly collisional rate increase between the $pr_{1L^{-1}}$ + SIP and $pr_{1L^{-1}}$ (without SIP) is five orders of magnitude while the difference between the INP and SIP simulations reaches six orders of magnitude. This increase in collisions increases the collisional breakup, especially between ice and snow and between small ice crystals, leading to a large contribution from this scheme (Fig. 11d).

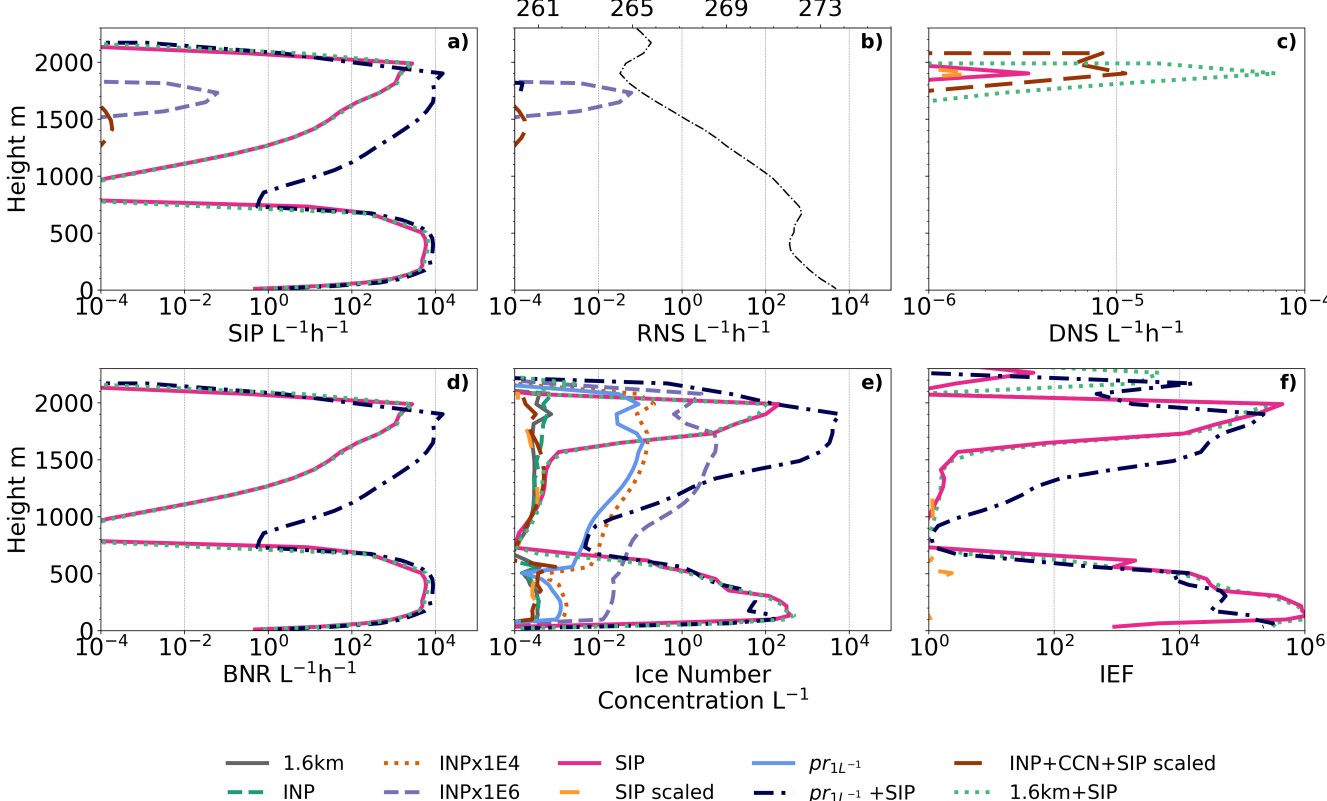

**Figure 11.** Average secondary ice production number concentration rates and ice number concentration shown for selected sensitivity studies on the 3rd of September 2020 between 06 UTC-12 UTC (before glaciation). (a) All SIP, (b) rime-splintering contribution (RNS), (c) droplet freezing and shattering (DNS), and (d) breakup upon ice-ice collisions (BNR). Panel (e) shows the ice number concentration used to calculate the ice enhancement factor (IEF) in panel (f). A temperature profile is added to panel (b) to distinguish the locations of the cloud layers. Please note the different ranges in the x-axis in panels c and f.

We find that breakup upon ice-ice collision is the dominating SIP. Previous modelling studies have deemed the contribution from the Takahashi et al. (1995) scheme too substantial and therefore introduce the concept of scaling this mechanism (Sotiropoulou et al., 2021; Georgakaki et al., 2022; Han et al., 2024). The scaled simulation (SIP scaled), whereby the breakup is scaled by the colliding particle diameter divided by the original particle diameter used in the experiments by Takahashi et al. (1995), nullifies the breakup contribution (Fig. 10 and contours in Fig. A1b). This might be due to very small frozen hydrometeor sizes that scale down this collisional rate too much or the lack of ice habit representation. A further study, using scaled breakup upon ice-ice collision, investigates whether scaling of the CCN concentration increases the droplet shattering mechanism (INP+CCN+SIP scaled). This simulation is also shown in Fig. 10 (contours are shown in Fig. A1d) and shows no large impact (on the frozen water path) from an increased droplet shattering with larger droplets (Fig. 11b). The fragments generated from this routine are simply too few to impact the cloud. Similarly to the $pr_{1L^{-1}}$ +CCN simulation, a decrease in

LWP is seen together with an increase in rain formation. A small rise in FWP can be seen in the lower layer (Fig. 10b) however, it is not substantial enough to impact the cloud phase further.

An ice enhancement factor, the increase in the number concentration of cloud ice due to SIP, can only be used for a constrained LES simulation where all ice within the domain has been formed during the simulation. Here, as we are dealing with a real setup, some of the cloud ice within the domain is supplied from the continuous advection of cloud hydrometeors through the domain boundaries. To gauge the impact of the SIP however, we may calculate the ratio between the simulations with and without SIP (apart from the rime-splintering which is included in all simulations but has a small impact) by $N_{SIP}/N_{Ref}$, where N is the number concentration of cloud ice and "SIP" and "Ref" refer to the simulation with and without SIP (with the same initial primary ice). We use time-averaged (06 UTC-12 UTC) ice number concentrations (shown in Fig. 11e) to gauge the impact of SIP and calculate the ratio between SIP implementations and their respective reference. The maximum ice number concentration enhancement ratio, shown vertically in Fig. 11f, amounts to $10^6$ for the lower INP simulations (1.6km+SIP vs 1.6km and SIP vs INP). For a larger initial INP concentration ($pr_{1L^{-1}}$) the maximum ice enhancement is $10^5$. This indicates a higher efficiency of the addition of SIP for a lower initial primary ice, as previously seen in the LES simulations by Sotiropoulou et al. (2020).

## 6   Discussion and conclusions

In this study, we investigate the microphysical sensitivity of Arctic mixed-phase multilayer clouds via the example of two cases observed during the MOSAiC campaign. Sensitivity simulations on cloud droplet activation, primary ice, and secondary ice processes were performed for two multilayer cloud systems in September 2020. The ICON model (Zängl et al., 2015) is used to model these cloud systems and observational data from the MOSAiC campaign (Shupe et al., 2022) is used to constrain and evaluate the model. The focus of this paper is two-fold, 1) to show that the thermodynamical structure of real cases of multilayer clouds can be (to a certain degree) accurately simulated providing a ground for further analysis into how these cloud systems form and evolve, and 2) to evaluate the microphysical sensitivities, in particular, the primary ice production and its impact on secondary ice production (SIP), while constraining the parameterisations to the observed ground-based measurements to better represent cloud phase.

Using the ICON model, a nested setup is adopted and through our analysis, we find that high-resolution (grid spacing finer than 1.6 km) simulations do not substantially improve the representation of the clouds. However, we acknowledge that finer resolutions than the 100 m studied here may be beneficial for many purposes (such as detailed studies of entrainment and turbulence, and especially idealised studies). We set up our model as a real case, with initial and boundary conditions from ICON Global 13 km analysis. The clouds are adequately represented in terms of vertical placement. However, during days when the local thermodynamic structure is not captured in the initial conditions, the model suffers from incorrectly placed cloud layers (vertical layering). The local structure and layering do not improve with higher resolution; these real-case setups suffer from carrying over large-scale biases in the initial and boundary conditions, which do not improve upon decreasing the grid spacing. Improvements in data assimilation may thus be crucial for accurately representing these clouds.

Switching from a 1-moment microphysics scheme to a 2-moment scheme facilitates comparisons with the real world, as the primary ice is represented using more explicit nucleation pathways. With the 2-moment microphysics and a 1.6 km grid spacing, we perform microphysical perturbations by constraining the model parameterisations with measured data. The model in its original form, severely underestimates the cloud ice mass concentration (by two orders of magnitude) while the cloud liquid water content is over-predicted by a factor of four. We perturb the aerosol parameterisations in an effort to 1) obtain the observed state, and 2) understand sensitivities in the cloud response to aerosols.

An immersion freezing parameterisation is created based on the surface-measured INPs to better represent the Arctic INP population. Accurate representation of these clouds is challenging and to adequately represent the observed ice mass concentration at warmer temperatures (T > -19°C), large scalings to the measurement-constrained immersion freezing parameterisation are needed, amounting to a factor between 1E4-1E6, which corresponds to prescribed INPs in the range of 1-100 L$^{-1}$. We hypothesise that the addition of the measured INPs is still too low to initiate further ice-phase processes, such as growth by deposition, and a larger scaling is required to get a response. The seeder-feeder mechanism, whereby frozen precipitation falls into a lower layer, was found to occur in the simulation with the highest INP concentration (INPx1E6). Falling snow from the lower layer increases by three orders of magnitude compared to reference simulations.

As the model produced too much liquid water, a CCN scaling of 0.1 was performed. The reduction of CCN concentration results in an expected rain enhancement due to the formation of larger droplets. This has no further impacts on the cloud ice due to the low rates of riming. With a prescribed CCN concentration and coarse horizontal grid spacing, capturing realistic cloud droplet activation is challenging. The saturation adjustment (condensation) in the model ensures that during supersaturated conditions (with respect to water), condensation occurs, making a reduction in cloud liquid difficult to obtain. The initialisation of cloud liquid (using 1-moment ICON at a coarser resolution) enables the saturation adjustment to act throughout the simulation (as long as nucleation still supplies newly activated droplets and/or cloud liquid is not entirely removed). Overall, cloud liquid is persistent. While it responds to changes in CCN and large perturbations of INPs, we find it difficult to reduce the modelled liquid water content to observed levels. This can potentially indicate a lack of efficiency in the WBF process as seen by previous studies (Omanovic et al., 2024) or a possible issue using a saturation adjustment scheme in a low CCN environment (Kogan and Martin, 1993). Prognostic aerosols using a dynamic aerosol model such as ICON-ART (Aerosol and Reactive Trace gases module) could potentially improve the representation of the local CCN concentrations and provide a more realistic cloud droplet activation. Furthermore, accurate representation of updrafts may ensure a more realistic simulation of cloud droplet activation, which could lead to an enhanced predictability of cloud liquid and layering. Improvements to the 1-moment scheme may further improve the cloud mass partitioning.

SIP has been investigated as these mechanisms are hypothesised to fill the gap in cloud ice number concentrations between models and observations. We find that the parameterisation for breakup upon ice-ice collisions (Sullivan et al., 2018) based on laboratory work by Takahashi et al. (1995) is very active, successfully glaciating the lower layer through the increase in small ice crystals and snow formation. This follows the findings by Sotiropolou et al. (2024) who show the dominance of collisional breakup for modelled Arctic clouds. The maximum enhancement in the cloud ice number concentration between simulations with and without SIP amounts to $10^5$-$10^6$, for a high and low initial INP concentration respectively. These factors are on the

larger side but are comparable to other Arctic studies that found ice enhancement up to $10^2$ (Sotiropoulou et al., 2020) and $10^4$ (Zhao et al., 2021). However, the enhancement from SIP is highly fluctuating with time, making a concrete analysis into whether or not the parameterisation is correctly representing the processes challenging. Scaling the breakup collision process by the particle diameter (Sotiropoulou et al., 2021; Georgakaki et al., 2022; Han et al., 2024) nullifies the SIP contribution. Further sensitivity studies, such as scaling the pre-factor in the parameterisation for breakup upon ice-ice collisions, as other studies have done (e.g. Sotiropoulou et al., 2020; Dedekind et al., 2021) may tune the collisional breakup rates to avoid the complete glaciation. Here, we will settle on the possibility that breakup may be a missing factor in these cloud layers and that we may not have the parameterisations available to properly explain the discrepancies. Droplet shattering was shown by Pasquier et al. (2022) to be the dominant SIP during their observational study on Svalbard. The negligible impact from the droplet shattering routine in this work may be due to the lack of adequate parameterisation. Improvements in the representation of cloud droplets and their freezing mechanisms might be crucial to better simulate the droplet freezing and shattering process.

The dependency on initial INP concentration alludes to the misrepresentation of the INPs through our measurement-constrained parameterisation and we conclude that the developed immersion freezing parameterisation is inadequate to capture the observed cloud ice mass concentration. We find that a large scaling of this immersion freezing (by 1E6) captures observed levels of ice, however, the required abundance of INPs in the high-Arctic is unrealistic. We conclude that secondary ice production combined with increased primary ice production is required to reach observed levels of ice while cloud liquid remains difficult to accurately capture. The sensitivities shown here also highlight the fact that these clouds respond very similarly to single-layer clouds.

In general, INP distributions are hard to capture with both models and with measurements (Burrows et al., 2022). Only ground-based measurements are available, which do not always correlate with INPs at cloud base (Creamean et al., 2021b), and we believe this may be part of the explanation. Another aspect is the lack of INP recycling in the model, whereupon INPs in the sub-cloud layer, deposited by precipitating ice, can re-enter the cloud and activate with possible time-dependent freezing rates, which is important for Arctic mixed-phase clouds (Solomon et al., 2015; Fu et al., 2019). INPs calculated from lidar measurements (Ansmann et al., 2023) may be the way forward and aerosol parameterisations based on their findings together with ground-based measurements may be a useful contribution to the modelling community. Through this exploration into the most accurate ways of representing these multilayer clouds, we hope to lay the foundation for further studies into how these layers form, evolve and impact the Arctic climate system.

*Data availability.* Full ICON model output is available on request from the authors. Data at the ship location will be published on the publicly available *RADAR4KIT* repository with the DOI:10.35097/hnu5eyk3xz4n3bkj.

## Appendix A: High resolution simulations

Clouds are sensitive to the resolution on which the simulation is done. Better representation can in general be expected when the grid spacing is decreased. Two simulations are performed on smaller grid spacing, which are initialised by the $pr_{1L-1}$ simulation. For the 400 m simulation (Fig. A1f), with a finer horizontal as well as vertical grid spacing (see Sect. 3), contours are similar to the $pr_{1L-1}$ simulation, Fig. A1c. The finest grid spacing simulation at 100 m changes the clouds marginally more than the 400 m setup and is shown for the 3rd in Fig. A1g. For the 100 m simulation, the explicit turbulence parameterisation is switched on and the vertical grid spacing is decreased. Most impacts seem to be due to the higher vertical resolution, rendering the cloud boundaries sharper. Some collisions are initiated giving a larger contribution of snow falling into the lower layer. This is, however, not enough to have any appreciable impact on the lower layer.

The time-averaged (06 UTC-12 UTC, Fig. A2b) maximum liquid water content in the lower layer barely increases between the simulations. In terms of droplet number concentrations larger impacts are seen with a 4-time (15-time) increase from the $pr_{1L-1}$ simulation for the 400 m (100 m) simulation respectively (Fig. A2a). Meanwhile, the cloud thickness decreases. With a smaller grid spacing an increase in vertical velocity is found which increases the CCN activation. An increase in cloud ice by a factor of 4 can be seen for the 100 m simulation (Fig. A2d), but this is deemed to be not enough impact to justify the large computational cost of performing microphysical sensitivity tests at such high resolution.

On the 1st (Fig. A3b,c), the high-resolution simulations impose sharper cloud boundaries, similar to the 3rd, and the upper cloud obtains more individual features due to the increased vertical resolution. The time-averaged (10 UTC-12 UTC) vertical profiles are shown in Fig. A4. An impact on cloud liquid is noted with a 3-time increase in droplet number concentration for the 100 m simulation compared to the $pr_{1L-1}$ simulation meanwhile the liquid water content only increases by a factor of 1.1. Cloud ice mass content in the upper layer increases by a factor of 2 while the ice number concentration increases by 5. The lower layer does not see an increase in cloud ice with a smaller horizontal grid spacing.

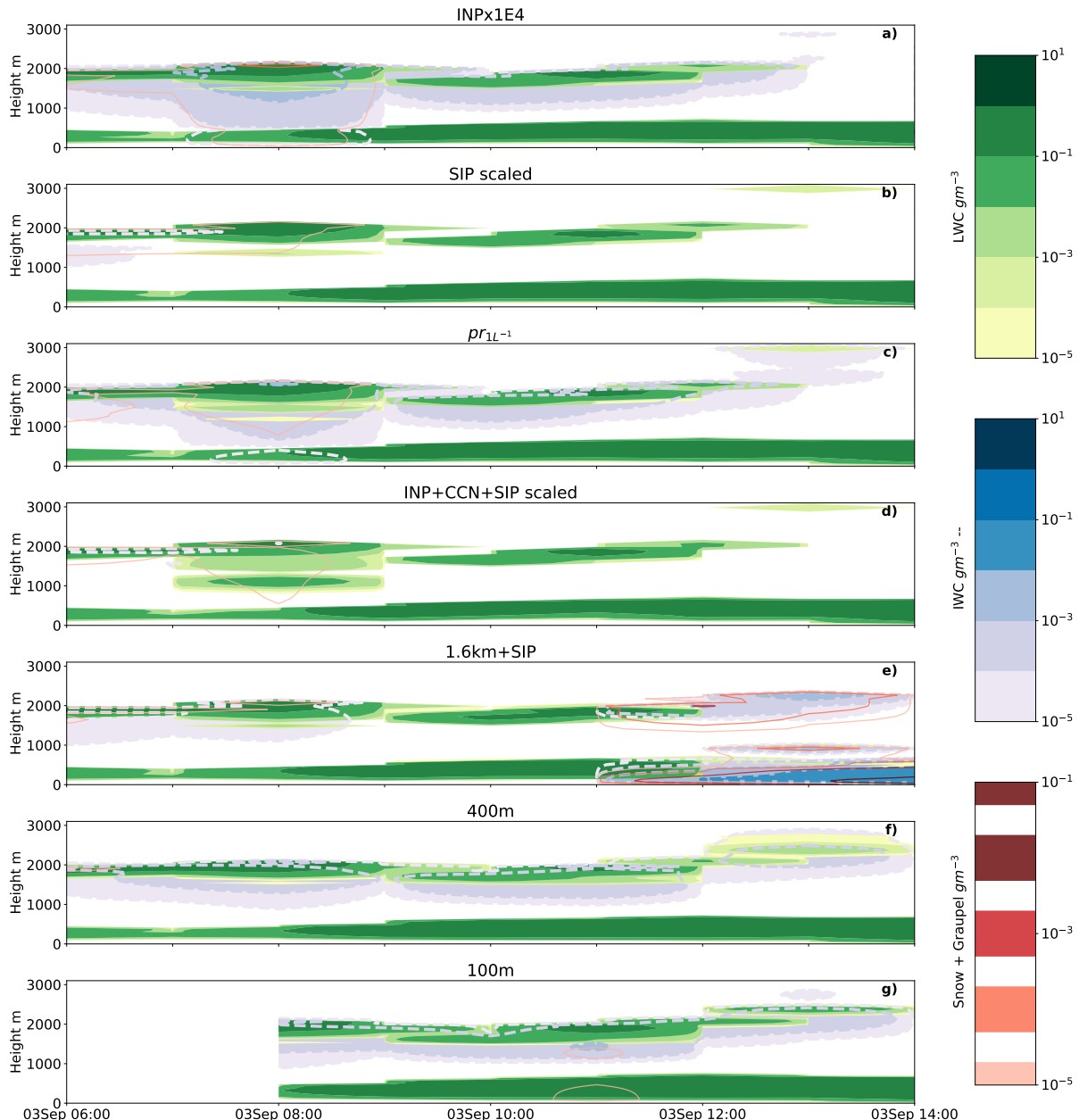

**Figure A1.** Time height plot of hydrometeors from the 3rd of September with liquid water mass content (green) and cloud ice water mass content (filled contours in blue and dashed outlines). (a) Polynomial scaling of 1E4, (b) SIP simulation with a scaled breakup upon ice-ice collision parameterisation, (c) the prescribed INP to 1 per L at warm temperatures, (d) new immersion freezing parameterisation with a CCN scaling by 0.1 and SIP included whereby the breakup upon ice-ice collisions is scaled down, (e) 1.6km+SIP, original primary ice production with added SIP, (f) 400 m simulation initialised using $pr_{1L^{-1}}$, and (g) 100 m simulation initialised using the 400 m simulation. Due to a 2 h delay between nests, the 100 m nest is only initialised at 08 UTC. Additional model parameters are the snow and graupel mass contents in red (solid line).

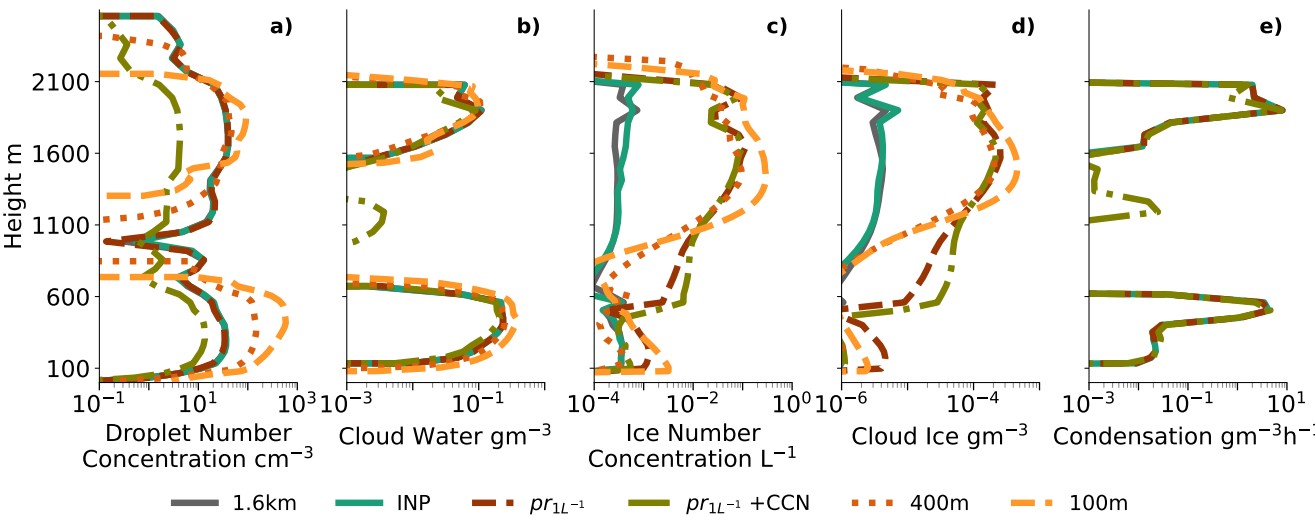

**Figure A2.** (a) Cloud droplet number concentrations, (b) cloud water mass content, (c) cloud ice number concentration, (d) cloud ice mass concentration and, first and second saturation adjustments, effectively the cloud droplet condensation routine, called to adjust the excess or deficit of vapour. Panel (d) only shows the first four simulations as listed in the legend. All were calculated for the 3rd of September as a temporal and spatial mean between 06 UTC-12 UTC.

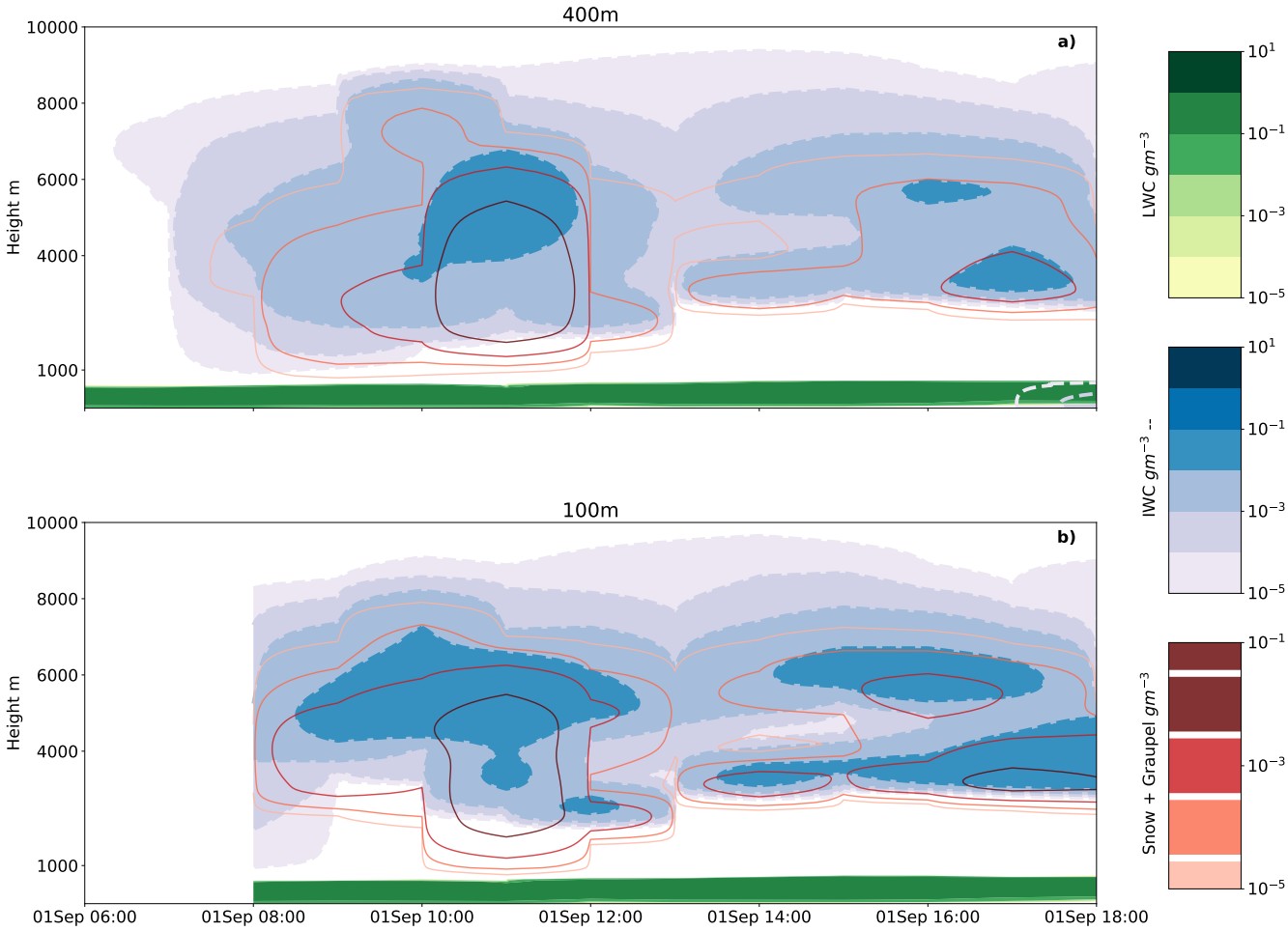

**Figure A3.** Time height plot of hydrometeors from the 1st of September with liquid water mass content (green) and cloud ice water mass content (filled contours in blue and dashed outlines). (a) 400 m simulation, initialised from the output of $pr_{1L-1}$ and, (b) 100 m initialised using the 400 m simulation. Due to a 2 h spin-up between nests, the 400 m (100 m) nest is only initialised at 06 UTC (08 UTC). Additional model parameters are the snow and graupel mass contents in red (solid line).

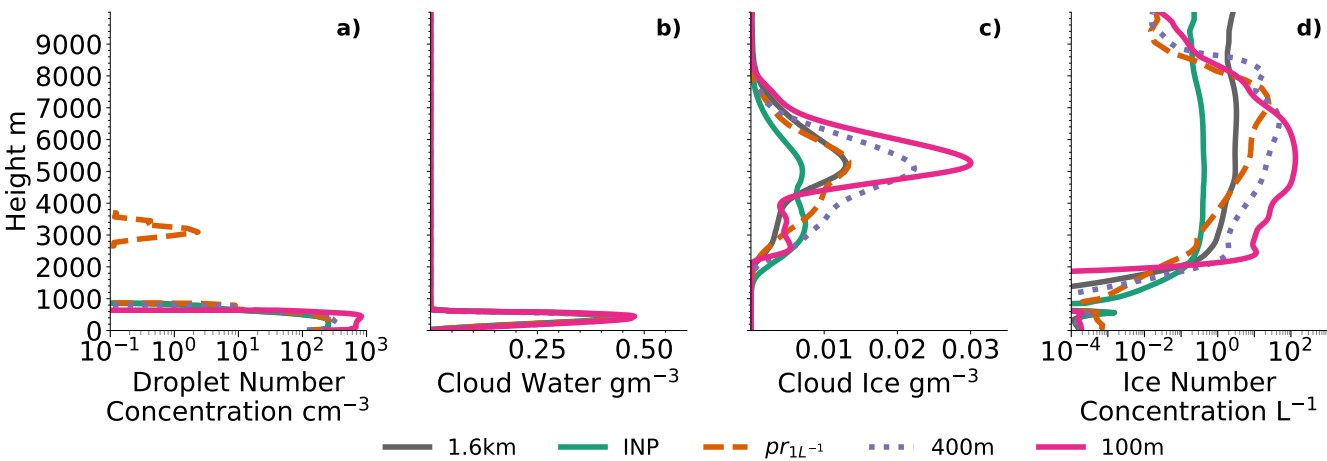

**Figure A4.** (a) Cloud droplet number concentrations, (b) cloud water mass content, (c) cloud ice mass content, and (d) cloud ice number concentration. All were calculated for the 1st of September as a temporal and spatial mean between 10 UTC-12 UTC.

**Appendix B**

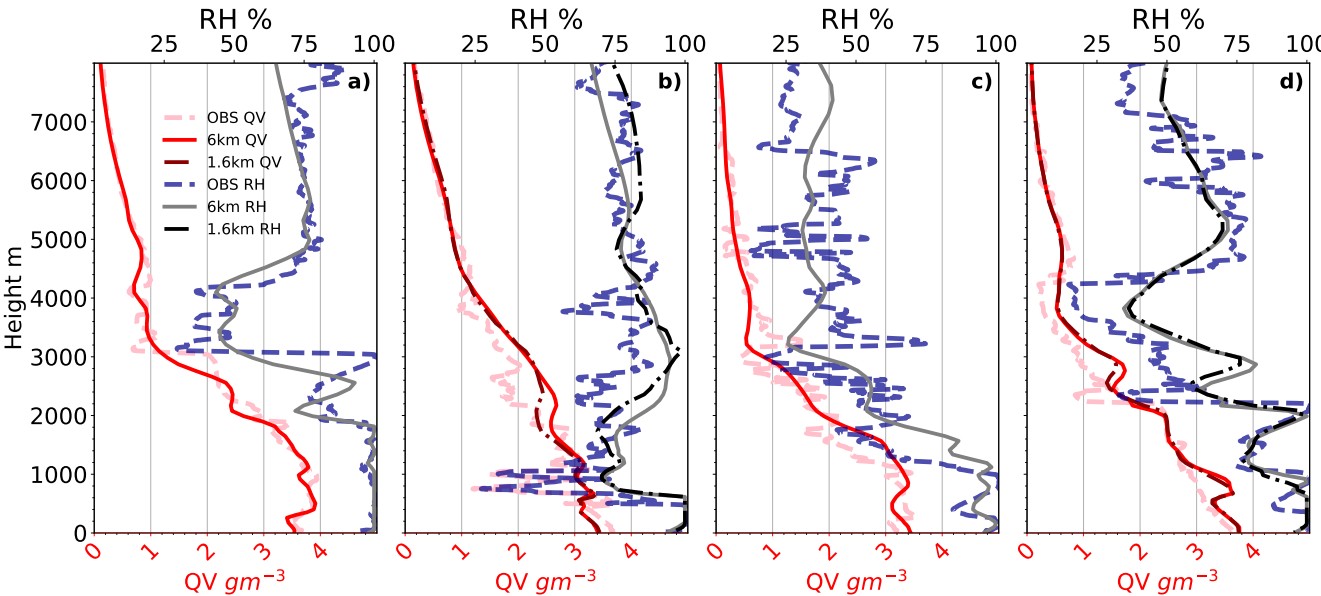

**Figure B1.** Vertical profiles of moisture content (QV) in red and relative humidity with respect to liquid water (RH) in grey. (a) 1st at initialisation time 00UTC, (b) 1st at 12UTC, (c) 3rd at initialisation time 00UTC, and (d) 3rd at 12UTC. The observational data (in dashed lines) are from the radiosondes (Maturilli et al., 2022) and the model data are the 6 km grid spacing simulation and at 12UTC for both days the 1.6 km simulation is additionally shown (initialisation at 04UTC).

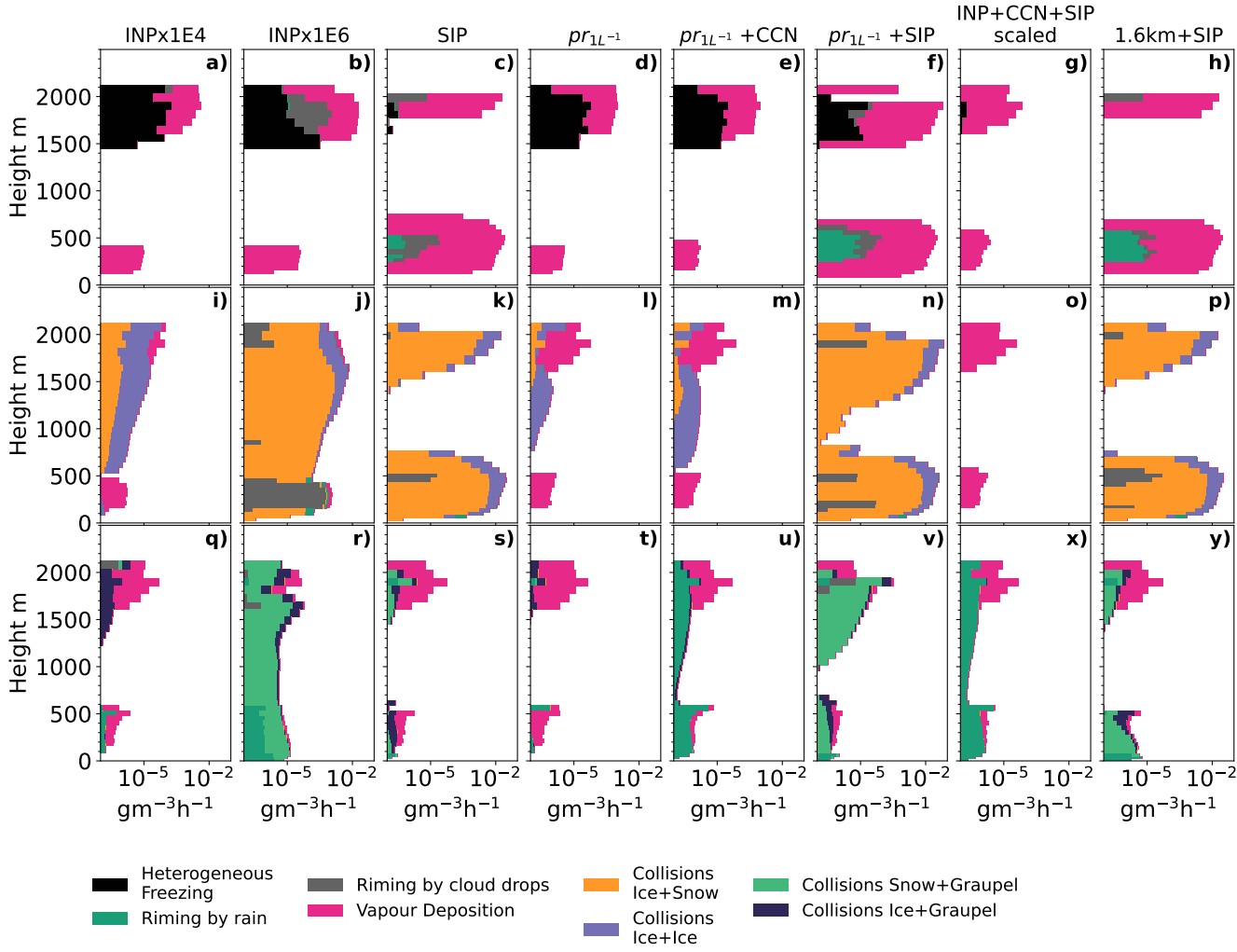

**Figure B2.** Microphysical process rates accumulated during the simulation and here taken as a mean between 06 UTC-12 UTC on the 3rd of September. a-h) Show processes contributing to ice mass growth, i-p) for snow and q-y) for graupel. Colours correspond to the various process rates defined in the legend.

*Author contributions.* GW devised and ran the simulations, wrote the manuscript and led the analysis. AO, LI, PA, MT, and CH provided input and discussions. AO ran the Lagranto trajectories and provided advice and discussions regarding the simulation setup. PA ran the MLC algorithm for the case selection. LI provided in-depth expertise and feedback to the manuscript. CH acquired funding and designed the project.

*Competing interests.* At least one of the (co-)authors is a member of the editorial board of Atmospheric Chemistry and Physics. The authors have no other competing interests to declare.

*Acknowledgements.* The authors would like to extend our gratitude to the anonymous reviewers who spent a considerable time to give feedback on the manuscript. The authors thank the Bundesministerium für Bildung und Forschung (BMBF) for funding the project with project number 03F0891B. We further thank all those who contributed to MOSAiC and made this endeavour possible (Nixdorf et al., 2021). We acknowledge ACTRIS and the Finnish Meteorological Institute for providing the observational data. This work was carried out on the HoreKa supercomputer funded by the Ministry of Science, Research and the Arts Baden-Württemberg and by the Federal Ministry of Education and Research. The authors would like to thank the Federal Ministry of Education and Research and the state governments (www.nhr-verein.de/unsere-partner) for supporting this project as part of the joint funding of National High-Performance Computing (NHR). This work was performed with the help of the Large Scale Data Facility at the Karlsruhe Institute of Technology funded by the Ministry of Science, Research and the Arts Baden-Württemberg and by the Federal Ministry of Education and Research. Grammarly and ChatGPT have been used for the improvement of the manuscript and plotting scripts. The author would also like to extend special thanks to Jessie Creamean and Matt Shupe for valuable discussions.

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
