# Peer review of "Sensitivities of Simulated Mixed-phase Arctic Multilayer Clouds to Primary and Secondary Ice Processes"

_EGUsphere, 2024_

## Referee Comment (RC1)

**Summary**

This paper focuses on improving the representation of multilayer mixed-phase clouds (MLCs) in the Arctic using the ICON model, a topic of significant importance given the critical role these clouds play in the Arctic climate system. By analyzing two case studies, the authors investigate the effects of various microphysical parameterizations and introduce a new immersion freezing parameterization based on their measurements. Their work aims to address the persistent challenges in simulating the mixed-phase character of these clouds, advancing our understanding of the factors influencing cloud formation and phase partitioning in the Arctic.

The study identifies key shortcomings in existing modeling approaches. Notably, the 1-moment microphysics scheme significantly underestimates ice concentrations, while the 2-moment scheme requires a dramatic increase in ice-nucleating particle (INP) concentrations - by a factor of $10^6$ - to match observed ice mass concentrations. Introducing secondary ice production (SIP) processes, such as ice-ice collision breakups, could provide a solution; however, scaling these processes realistically diminishes their impact on ice crystal numbers, suggesting potential gaps in the current parameterizations.

Additionally, the authors find that increasing grid resolution beyond 1.6 km does not improve simulation accuracy, as large-scale biases dominate over resolution effects. Similarly, reductions in cloud condensation nuclei (CCN) concentrations yield negligible effects. These findings underscore the need for improved SIP representations to enhance future Arctic ice simulations.

**General comments**

**Assessment of the new immersion freezing parametrization**

The authors do not clearly state whether they recommend the general use of the new immersion freezing parameterization in the Arctic or discuss its quantitative impact on SIP. A setup testing the magnitude of SIP with the original INP parameterization is missing. Including such a setup would help clarify whether the introduction of the new immersion freezing parameterization (without scaling) alters the impact of SIP, or if an accurate representation of INPs is not essential for modeling MLCs under the conditions studied.

**Revisiting the role of INPs and the importance of SIP**

The conclusion that warm-temperature INPs are crucial for modeling MLCs is not convincingly demonstrated, as this is only observed under unrealistically high INP concentrations (e.g., 1E4 and 1E6). Instead, the results suggest that SIP may be the dominant missing factor.

**Critical discussion of SIP processes**

While the focus on SIP's role in phase partitioning is valuable, the study lacks critical discussions that would provide a more balanced perspective:

1. **Ice-Ice Collisions**: The breakup parameterization for ice-ice collisions only shows an effect in its original version, which is acknowledged to be unrealistic. The implications of this limitation should be explored more thoroughly.
2. **Droplet Shattering:** This process is mentioned briefly but dismissed as not very relevant. However, the original parameterization is based on limited measurements and may underestimate its potential impact. Recent studies (e.g., (Lawson et al. 2023)) suggest that droplet shattering (DS) could be significant in the Arctic, particularly at higher temperatures. Could the poor representation of DS in the model mask its true importance in explaining observed ice concentrations?
3. **Cloud condensation nuclei scaling and SIP**: The study finds that scaling cloud condensation nuclei (CCN) to match observations increases rainfall but has minimal other effects. Yet, larger droplets from scaled CCN could enhance DS, potentially influencing SIP. This connection deserves further exploration.

**Summary**

In summary, while the study contributes to understanding Arctic MLCs, it would benefit from:
- Clarification on the general applicability of the new immersion freezing parameterization.
- More rigorous analysis of SIP processes and parameterizations, especially the role of DS.

These additions would enhance the paper's impact and provide a more comprehensive understanding of the factors controlling phase partitioning in Arctic clouds.

**Specific comments**

**Clarification of ice metrics**

It is unclear at times whether the authors are discussing ice crystal number concentration or ice mass concentration. In the abstract, the INP concentration is stated to need an increase by a factor of $10^6$ to match observed ice mass concentration, while the ice-ice collision breakup increases cloud ice number concentration by a factor of $10^5$. These two metrics are not directly comparable. Is SIP increasing ice crystal number concentration, ice mass concentration, or both? Please ensure consistent and precise terminology throughout the manuscript.

**Inconsistent modeling approaches across case studies**

The authors use a wide variety of setups and parameterizations for the third case study but do not apply a similar approach to the first case study, despite both suffering from the same fundamental issue - the lack of a mixed-phase character in the base experiment. Why weren't additional setups tested for the first case study? None of the setups used for the first case resolves this issue, and further exploration seems warranted.

**Terminology for INP scaling**

The term *INP perturbation* is somewhat confusing. Would "scaled INP" be a clearer and more accurate description? Additionally, the terminology is inconsistently applied; at times, it is referred to as *tuned* or *polynomial scaled*, as seen in Figure 1. I recommend consistently using "scaled INP" throughout the paper for clarity and uniformity.

**Abbreviations for scaled warm INP**

For the polynomial-scaled warm INP, the authors sometimes use terms like $INPx10^4$ or *1E4* but at other times refer to it more generally as *perturbed INP concentration*. To maintain consistency, I suggest introducing and consistently using abbreviations such as *1E4* and *1E6*. The introduction could be done in Section 3.1.2, "Primary Cloud Ice Formation". This would align with the introduction of the scaled CCN parametrization in Section 3.1.1. This would improve clarity and make it easier for readers to follow the paper's content.

**Underuse of the second case study**

The data for the second case study is minimally utilized in the paper, with only a couple of sentences dedicated to it. If it does not contribute substantially to the conclusions, consider removing it entirely and focusing on the first and third case studies. This would improve the paper's focus and streamline the discussion.

**Specification of model and schemes**

The manuscript does not specify which version of ICON was used or whether the implicit or explicit two-moment microphysics scheme was employed. Given that the implicit scheme is the default, it is likely the one used, but this should be clearly stated in the text to avoid ambiguity.

**Public accessibility of data**

To support transparency and reproducibility, the data used in this study should be made publicly available. This is a crucial step for enabling further research and validation.

**Technical corrections**

1. **Line 46**: The term "successfully" seems overly strong in this context. In contrast, Korolev and Leisner (2020) emphasize the need for further laboratory studies to obtain a quantitative understanding of the efficiencies of individual SIP mechanisms. Consider rephrasing to reflect the ongoing challenges and uncertainties in this area.

2. **Line 57**: The phrase "cloud ice concentration" should be replaced with "cloud ice number concentration" for precision. As written, "cloud ice concentration" could refer to either number concentration or mass concentration, which are distinct quantities, and the sentence would not hold true for mass concentration. Apart from this, there are several other sentences in the paper, where it is not clear, if the authors talk about number or mass concentration.

3. **Lines 173–174**: The description of droplet shattering is misleading. It refers to the process where large supercooled droplets (typically >50 μm in diameter) fragment into smaller ice particles, *not* the freezing of rain droplets. Refer to Korolev and Leisner (2020) for more accurate details.

4. **Lines 230–233**: This sentence is overly complex and difficult to follow. Please split it into two sentences for better readability and understanding.

5. **Line 264 onward**: The authors mention an inversion predicted at 2 km, 1 km higher than observed. However, Figure 6d also shows an inversion predicted at 600 m, which is not discussed in the evaluation. This omission should be addressed.

6. **Line 336**: The impact of the CCN scaling is smaller compared to what? Please clarify by providing a complete sentence to specify the reference point for this comparison.

7. **Line 360**: The section title "Can the INP perturbation be explained by SIP" seems misleading. The term "INP perturbation" refers to an artificial adjustment introduced in the study rather than a physical phenomenon. Given the author'ss finding that INP concentrations would need to increase by a factor of $10^6$ to match observations, a more appropriate question might be whether SIP can instead account for the discrepancy between modeled INP concentrations and observed values. Consider revising the title to reflect this more accurately.

8. **Lines 399–400**: The mention of the maximum ice number concentration enhancement ratio is unclear. Is this information presented in any figure? Figure 9 only shows ice water content, which does not provide insight into ice number concentrations.

9. **Line 400:** The abbreviation "QNI" is used but never introduced or defined. Please provide its meaning when it first appears.

10. **Figure 1**: To show all the evaluated setups in your study, consider including $pr_{1L-1}$ in the figure, even if it only adds a single line to the plot. This would provide a more comprehensive overview of the setups analyzed.

11. **Figure 3**: Please clarify the unit in the figure. Are you referring to the sea surface ice fraction?

12. **Figure 4**:
- The base and cloud tops do not appear to align with the measurements of ice water content. Could you explain this discrepancy?
- Additionally, there seems to be a missing cloud top at 00:00 UTC on September 2. Please verify and address this.

13. **Figure 5**:
- Is this figure based on model output or measurements? Please clarify in the caption.
- The mean sea level pressure is difficult to read, and the ship location (marked with a star) is barely visible. Consider increasing the plot size and improving the visibility of key elements.

14. **Figure 10**: The markers for "SIP scaled", "1.6 km", and "INP" setups overlap, making the comparisons discussed in the text difficult to distinguish. Could you consider alternative visualization methods, such as using smaller markers or providing a zoomed-in view of this section? Additionally, are precipitation observations available that could be included for reference to further validate the model's performance?

15. **Section 4.1 Synoptic Situation**: The detailed description of back trajectories would be more effective if accompanied by the corresponding figure in the main text rather than in the appendix. This would improve the flow and understanding of the discussion.

**References**

Alexei V. Korolev and Thomas Leisner, 2020. *Review on experimental studies of secondary production*. Vol. 20, 11767-11797, Atmos. Chem. Phys.

R. Paul Lawson, Alexei V. Korolev, Paul J. DeMott, Andrew J. Heymsfield, Roelof T. Bruintjes, Cory A. Wolff, Sarah Woods, Ryan J. Patnaude, Jørgen B. Jensen, Kathryn A. Moore, Ivan Heckman, Elise Rosky, Julie Haggerty, Russell J. Perkins, Ted Fisher and Hill, Thomas. C. J., 2023. *The Secondary Production of Ice in Cumulus Experiment (SPICULE)*. Vol. 104(1), E51-E76, BAMS.

---

## Author Response (AR1)

**Author's response to reviewer 1**

We thank reviewer #1 for their detailed and constructive feedback on the submitted paper. We have responded to each comment in detail below. A simple "Ok" means we have changed it, for larger changes, the sentence or figure in question has been included. Figures shown in this document are labelled using Roman numerals and referrals to the original figures are included when appropriate.

**Review 1**

**Summary**

This paper focuses on improving the representation of multilayer mixed-phase clouds (MLCs) in the Arctic using the ICON model, a topic of significant importance given the critical role these clouds play in the Arctic climate system. By analyzing two case studies, the authors investigate the effects of various microphysical parameterizations and introduce a new immersion freezing parameterization based on their measurements. Their work aims to address the persistent challenges in simulating the mixed-phase character of these clouds, advancing our understanding of the factors influencing cloud formation and phase partitioning in the Arctic. The study identifies key shortcomings in existing modeling approaches. Notably, the 1-moment microphysics scheme significantly underestimates ice concentrations, while the 2-moment scheme requires a dramatic increase in ice-nucleating particle (INP) concentrations - by a factor of $10^6$ - to match observed ice mass concentrations. Introducing secondary ice production (SIP) processes, such as ice-ice collision breakups, could provide a solution; however, scaling these processes realistically diminishes their impact on ice crystal numbers, suggesting potential gaps in the current parameterizations.

Additionally, the authors find that increasing grid resolution beyond 1.6 km does not improve simulation accuracy, as large-scale biases dominate over resolution effects. Similarly, reductions in cloud condensation nuclei (CCN) concentrations yield negligible effects. These findings underscore the need for improved SIP representations to enhance future Arctic ice simulations.

**General comments**

**Assessment of the new immersion freezing parametrization**

The authors do not clearly state whether they recommend the general use of the new immersion freezing parameterization in the Arctic or discuss its quantitative impact on SIP.

→ We believe it is hard to justify the use of this new parameterisation in general as the impact of the parameterisation alone is almost negligible. However, we'd like to stress the need for it for further investigations of primary and secondary ice production to obtain the observed state. We add a clarification statement in the discussion:
"The dependency on initial INP concentration alludes to the misrepresentation of the INPs through our measurement-constrained parameterisation and we conclude that the developed immersion freezing parameterisation is not enough to capture the observed cloud ice mass concentration."

A setup testing the magnitude of SIP with the original INP parameterization is missing. Including such a setup would help clarify whether the introduction of the new immersion freezing parameterization (without scaling) alters the impact of SIP, or if an accurate representation of INPs is not essential for modeling MLCs under the conditions studied.

→ Yes, you are quite right. We have thus performed a new simulation using the original parameterisations for primary ice production together with SIP (1.6km+SIP). Process rates such as shown in Fig. B5 in the original manuscript are shown below (Fig.i). The new simulation (1.6km+SIP) is shown in green. Fig. i is now added to the main text as Fig. 11.

[Figure]

Fig. i: Updates to Fig.B6 now included in the main text as Fig. 11. Two new runs have been added; INP+CCN+SIP scaled and 1.6km+SIP and ice enhancement factor (f) (see Line 399).

An increase in the droplet shattering process rates for the 1.6km+SIP can be seen compared to the SIP setup. This is due to the lack of constraint on the rain freeze as a primary ice production mechanism. However, this has almost no impact on the total mass production from SIP (Fig.i a) nor the total ice number concentration (Fig.i e). However, we see here clearly that the SIP is dominating at these low immersion nuclei concentrations. Figure 10 (scatter plot of FWP and LWP) in the original

manuscript is updated accordingly and is shown below (Fig. ii). We find that SIP is not enough to reach observed levels of ice on its own but requires an INP scaling at warm temperatures (pr_1L$^{-1}$ +SIP).

**Revisiting the role of INPs and the importance of SIP**

The conclusion that warm-temperature INPs are crucial for modeling MLCs is not convincingly demonstrated, as this is only observed under unrealistically high INP concentrations (e.g., 1E4 and 1E6). Instead, the results suggest that SIP may be the dominant missing factor.

→ See the comment above, both are required for the most accurate/realistic depiction of the clouds in terms of their integrated water paths. The pr_1L$^{-1}$ simulation produces a similar amount of ice mass concentration as the SIP simulation (based on the low INP concentration) while the combination of the two reaches values above the observed. The effect of SIP is found to be larger for a smaller initial INP concentration and thus the addition of SIP on the pr_1L$^{-1}$ simulation has a smaller impact. Due to these conclusions, we argue that our initial INP concentration, measured at the ground is likely not representative of in-cloud conditions. Following this argument, a higher initial INP concentration would give a smaller relative impact from SIP and thus we conclude that the INPs are critical to better capture the clouds. However, SIP is likely present, but the large impact seen here, as we discuss in the discussion section, is unphysical due to the badly constrained parameterisation.

**Critical discussion of SIP processes**

While the focus on SIP's role in phase partitioning is valuable, the study lacks critical discussions that would provide a more balanced perspective:

Ice-Ice Collisions: The breakup parameterization for ice-ice collisions only shows an effect in its original version, which is acknowledged to be unrealistic. The implications of this limitation should be explored more thoroughly.

→ A scaled version of the breakup upon ice-ice collision parameterisation is also provided in section 5.4 following other papers in the area of breakup collision impacts. Arbitrarily tuning the prefactors was not deemed to be worth the computing time as a decrease in the SIP effect then requires a larger INP scaling. Thus, tuning this parameterisation to fit the data would not provide us with any reliable outcomes in terms of how realistic the parameterisation is.

 Droplet Shattering: This process is mentioned briefly but dismissed as not very relevant. However, the original parameterization is based on limited measurements and may underestimate its potential impact. Recent studies (e.g., (Lawson et al.

2023)) suggest that droplet shattering (DS) could be significant in the Arctic, particularly at higher temperatures. Could the poor representation of DS in the model mask its true importance in explaining observed ice concentrations?

→ This is of course a limitation all parameterisations face. Through your comments, we have realised we may not have made it clear enough how droplet shattering is parameterised.

We have now added to Sect. 3.1.3: "This process is parameterised within the primary ice production parameterisation for rain (and drizzle) freeze and will inherently depend on the rate of rain freeze."

Added to Sect. 3.2: "To be emphasised here, the droplet shattering mechanism is tuned down due to the tuning of the rain freeze mechanism, where the droplet shattering is implemented."

As shown in Fig. i above, droplet shattering increases when the rain (and drizzle mechanism) is not tuned down (1.6km+SIP), however, its impact on ice mass and number concentrations is still negligible.

We have now added a few sentences about the discrepancy towards the observed SIP such as Pasquier et al. 2022 who also see a larger DS occurrence in Arctic clouds. This citation may be more appropriate than the Lawson et al. 2023 paper (we cannot find a mention of the Arctic in the paper provided).

These sentences are added to the discussion:

" Droplet shattering was shown by Pasquier et al. 2022 to be the dominant SIP during their observational study on Svalbard. The negligible impact from the droplet shattering routine in this work may be due to the lack of adequate parameterisation. Improvements in the representation of cloud droplets and their freezing mechanism might be crucial to better simulate the droplet freezing and shattering process."

Cloud condensation nuclei scaling and SIP: The study finds that scaling cloud condensation nuclei (CCN) to match observations increases rainfall but has minimal other effects. Yet, larger droplets from scaled CCN could enhance DS, potentially influencing SIP. This connection deserves further exploration.

→ In addition to the simulations shown in the paper, we have a simulation performed with scaled SIP (scaled-down breakup upon ice-ice collisions) and scaled-down CCN (by 0.1) with the new parameterisation for immersion freezing. We have now added this simulation to the paper and below you can see the updated figure (Fig. ii) as well as the figure showing process rates above (Fig. i). This simulation is shown as "INP+CCN+scaled SIP" and shows no large impact (on the frozen water path) from an increased droplet shattering with larger droplets (Fig.i above). A small increase is seen in the lower layer but only a marginal difference.

[Figure]

Fig. ii: Updates to Fig. 10 in the original manuscript with the two new simulations added (INP+CCN+SIP scaled, 1.6km+SIP) and updates to the marker sizes.

We add these sentences:

"A further study using the scaled breakup upon ice-ice collision investigates whether scaling of the CCN concentration has an impact on the droplet shattering mechanism (INP+CCN+SIP scaled). This simulation is also shown in Fig. 10 (contours are shown in Fig.A1d) and shows no large impact (on the frozen water path) from an increased droplet shattering with larger droplets. Similarly to the $pr_{1L}^{-1}$+CCN simulation, a small decrease in LWP is seen together with an increase in rain formation. A small rise in FWP can be seen in the lower layer (Fig.10b) however, not substantial enough to impact the cloud phase further."

**Summary**

In summary, while the study contributes to understanding Arctic MLCs, it would benefit from:

- Clarification on the general applicability of the new immersion freezing parameterization.

- More rigorous analysis of SIP processes and parameterizations, especially the role of DS.

→ Thank you for your constructive comments. We have addressed your questions and remarks in the revised version of the manuscript. Detailed replies to your general comments have been provided above.

These additions would enhance the paper's impact and provide a more comprehensive understanding of the factors controlling phase partitioning in Arctic clouds.

**Specific comments**

**Clarification of ice metrics**

It is unclear at times whether the authors are discussing ice crystal number concentration or ice mass concentration. In the abstract, the INP concentration is stated to need an increase by a factor of $10^6$ to match observed ice mass concentration, while the ice-ice collision breakup increases cloud ice number concentration by a factor of $10^5$. These two metrics are not directly comparable. Is SIP increasing ice crystal number concentration, ice mass concentration, or both? Please ensure consistent and precise terminology throughout the manuscript.

→ Thank you for bringing this up, we have now edited all instances where it is unclear.

**Inconsistent modeling approaches across case studies**

The authors use a wide variety of setups and parameterizations for the third case study but do not apply a similar approach to the first case study, despite both suffering from the same fundamental issue - the lack of a mixed-phase character in the base experiment. Why weren't additional setups tested for the first case study? None of the setups used for the first case resolves this issue, and further exploration seems warranted.

→ The first case study is too warm in the modelled lower layer cloud (see Figure 6b in the original manuscript) for any ice nucleation to occur and thus we deemed it unnecessary to further explore primary and secondary ice production within this lower layer.

We add a sentence clarifying the lack of further studies:
"The upper layer shows only small changes due to the scaling of the heterogeneous parameterisations while the lower layer is unaffected by the changes due to its high temperature. Further studies on primary and secondary ice processes in the lower layer are deemed unnecessary. "

**Terminology for INP scaling**

The term INP perturbation is somewhat confusing. Would "scaled INP" be a clearer and more accurate description? Additionally, the terminology is inconsistently applied; at times, it is referred to as tuned or polynomial scaled, as seen in Figure 1. I recommend consistently using "scaled INP" throughout the paper for clarity and uniformity.

→ Thank you for pointing this out. The words "tuned" and "perturbation" have been replaced by "scaled" where it is appropriate

**Abbreviations for scaled warm INP**

For the polynomial-scaled warm INP, the authors sometimes use terms like INPx104 or 1E4 but at other times refer to it more generally as perturbed INP concentration. To maintain consistency, I suggest introducing and consistently using abbreviations such as 1E4 and 1E6. The introduction could be done in Section 3.1.2, "Primary Cloud Ice Formation". This would align with the introduction of the scaled CCN parametrization in Section 3.1.1. This would improve clarity and make it easier for readers to follow the paper's content.

→ Thank you for bringing this up, we realise we did not fully introduce these two perturbations. We have added in Section 3.2:

 "A factor of 1E4 and 1E6 is applied to the polynomial, increasing the immersion freezing at temperatures between -20C<T<-7C. "

Furthermore, we have changed instances of $10^x$ to 1EX, when referring to the simulations.

**Underuse of the second case study**

The data for the second case study is minimally utilized in the paper, with only a couple of sentences dedicated to it. If it does not contribute substantially to the conclusions, consider removing it entirely and focusing on the first and third case studies. This would improve the paper's focus and streamline the discussion.

→ We wanted to include this day for the overall view of these 3 days. Further studies have been performed on the 2nd of September which will be the focus of future publications, and thus, an introduction to the day seemed warranted.

**Specification of model and schemes**

The manuscript does not specify which version of ICON was used or whether the implicit or explicit two-moment microphysics scheme was employed. Given that the

implicit scheme is the default, it is likely the one used, but this should be clearly stated in the text to avoid ambiguity.

→ Thank you for pointing this out, we add the sentence:

"We use ICON version 2.6.5 with a semi-implicit time integration solver."

**Public accessibility of data**

To support transparency and reproducibility, the data used in this study should be made publicly available. This is a crucial step for enabling further research and validation.

→ All data can be made available upon request. As it is very large (approximately 13TB) we are unable to publish all of it, however, we have decided to publish the data from the smaller evaluation region on the publicly available *KITopen* repository before the final paper is submitted.

**Technical corrections**

Line 46: The term "successfully" seems overly strong in this context. In contrast, Korolev and Leisner (2020) emphasize the need for further laboratory studies to obtain a quantitative understanding of the efficiencies of individual SIP mechanisms. Consider rephrasing to reflect the ongoing challenges and uncertainties in this area.

→ Removed "successfully". The end of the sentence ("but the atmospheric relevance remains unclear.") hopefully reflects the uncertainty enough.

Line 57: The phrase "cloud ice concentration" should be replaced with "cloud ice number concentration" for precision. As written, "cloud ice concentration" could refer to either number concentration or mass concentration, which are distinct quantities, and the sentence would not hold true for mass concentration. Apart from this, there are several other sentences in the paper, where it is not clear, if the authors talk about number or mass concentration.

→ Thank you for pointing this out. We have now corrected all instances of "ice concentration" to the appropriate one.

Lines 173–174: The description of droplet shattering is misleading. It refers to the process where large supercooled droplets (typically >50 μm in diameter) fragment into smaller ice particles, not the freezing of rain droplets. Refer to Korolev and Leisner (2020) for more accurate details.

→ The reason "rain" has been used is due to its implementation in the code, where droplets of the relevant sizes fall into the drizzle and raindrop category. However, we can understand the ambiguity of this and we have now changed "rain droplets" to "large supercooled cloud droplets" in this statement as well as in line 54 to avoid confusion.

Lines 230–233: This sentence is overly complex and difficult to follow. Please split it into two sentences for better readability and understanding.

→ Done

Line 264 onward: The authors mention an inversion predicted at 2 km, 1 km higher than observed. However, Figure 6d also shows an inversion predicted at 600 m, which is not discussed in the evaluation. This omission should be addressed.

→ This is mentioned in Line 257, however inaccurately stated as 500m instead of 600m, thank you for noticing.

Line 336: The impact of the CCN scaling is smaller compared to what? Please clarify by providing a complete sentence to specify the reference point for this comparison.

→ Changed "smaller" to "small" as it is not a comparison.

Line 360: The section title "Can the INP perturbation be explained by SIP" seems misleading. The term "INP perturbation" refers to an artificial adjustment introduced in the study rather than a physical phenomenon. Given the author's finding that INP concentrations would need to increase by a factor of $10^6$ to match observations, a more appropriate question might be whether SIP can instead account for the discrepancy between modeled INP concentrations and observed values. Consider revising the title to reflect this more accurately.

→ We have changed it to "Can the lack of cloud ice be explained by SIP?"

Lines 399–400: The mention of the maximum ice number concentration enhancement ratio is unclear. Is this information presented in any figure? Figure 9 only shows ice water content, which does not provide insight into ice number concentrations.

→ We have now changed our approach to use time-averaged ice number concentrations, that show the same enhancement. The time-averaged ice number concentrations are shown in Fig. i above. We also add the ice enhancement factor as a vertical profile in Fig.i f(above).

The sentences now read:

"We use maximum time-averaged (06-12UTC) ice number concentrations (shown in Fig.11e) to gauge the impact of SIP and calculate the ratio between SIP implementations and their respective reference. The average ice number concentration enhancement ratio, shown vertically in Fig. 11f, amounts to 10^5 for the lower INP simulations (1.6km+SIP vs 1.6km and SIP vs INP). For a larger initial INP concentration (pr_1L$^{-1}$) the enhancement is 10^4."

Line 400: The abbreviation "QNI" is used but never introduced or defined. Please provide its meaning when it first appears.

→ Thank you for noticing, we have changed it in the x-axis label.

Figure 1: To show all the evaluated setups in your study, consider including pr 1L-1 in the figure, even if it only adds a single line to the plot. This would provide a more comprehensive overview of the setups analyzed.

→ Thank you for pointing this out, we have now changed the figure (see Fig. iii below).

[Figure]

Fig. iii: Updates to Fig. 1 in the original manuscript with an added second x-axis in Celsius and the INP constraint of 1 per L (pr_1L$^{-1}$) added.

Figure 3: Please clarify the unit in the figure. Are you referring to the sea surface ice fraction?

→ The figure caption states that it is sea ice concentration, for clarity we have added "Sea Ice Concentration" to the colour bar label.

Figure 4:

- The base and cloud tops do not appear to align with the measurements of ice water content. Could you explain this discrepancy?

→ We have added the sentence "Discrepancies may be due to drifting radiosondes and precipitation (falling ice crystals), which complicates the determination of cloud boundaries."

- Additionally, there seems to be a missing cloud top at 00:00 UTC on September. Please verify and address this.

→ Thank you for noticing. This cloud top is above the plotted area (>8.5km). This is now stated in the caption.

Figure 5:

- Is this figure based on model output or measurements? Please clarify in the caption.

→ Changed to "Modelled synoptic situation" in the figure caption.

- The mean sea level pressure is difficult to read, and the ship location (marked with a star) is barely visible. Consider increasing the plot size and improving the visibility of key elements.

→We have increased the size of the star as well as the contour labels, we hope this is large enough to read. An updated plot, together with trajectories can be seen below (Fig. iv)

[Figure]

Fig. iv: (a) Synoptic situation similar to Fig.3 in the original manuscript and (b) back trajectories similar to Fig. B1 in the original manuscript.

Figure 10: The markers for "SIP scaled", "1.6 km", and "INP" setups overlap, making the comparisons discussed in the text difficult to distinguish. Could you consider alternative visualization methods, such as using smaller markers or providing a zoomed-in view of this section? Additionally, are precipitation observations available that could be included for reference to further validate the model's performance?

→ We have updated the plot with markers in different sizes, thank you for the suggestions. The plot is shown above (Fig.ii) under the section "Critical discussion of SIP processes". We have available laser disdrometer data, however, these would not be comparable to the rain category in this plot as this is rain within the column and not precipitation at the ground.

Section 4.1 Synoptic Situation: The detailed description of back trajectories would be more effective if accompanied by the corresponding figure in the main text rather

than in the appendix. This would improve the flow and understanding of the discussion.

→This is now plotted on the side of the synoptic situation and included in the main manuscript (see Fig. iv).

References

Alexei V. Korolev and Thomas Leisner, 2020. Review on experimental studies of

secondary production. Vol. 20, 11767-11797, Atmos. Chem. Phys.

R. Paul Lawson, Alexei V. Korolev, Paul J. DeMott, Andrew J. Heymsfield, Roelof T. Bruintjes, Cory A. WolG, Sarah Woods, Ryan J. Patnaude, Jørgen B. Jensen, Kathryn A. Moore, Ivan Heckman, Elise Rosky, Julie Haggerty, Russell J. Perkins, Ted Fisher and Hill, Thomas. C. J., 2023. The Secondary Production of Ice in Cumulus Experiment (SPICULE). Vol. 104(1), E51-E76, BAMS.

Additional changes:

New information regarding the data assimilation strategy of MOSAiC radiosondes has been obtained. All radiosondes were assimilated from the campaign. Variables include temperature, wind profiles, and relative humidity.  We are updating the manuscript accordingly.

**Review 2**

This manuscript addresses the topic of Arctic multilayer clouds, and specifically ice related processes, using a nested modeling approach and case studies from the MOSAiC expedition. Broadly the topic of Arctic clouds, and mixed-phase clouds specifically, is an important one because of ongoing modeling challenges in representing Arctic cloud phase partitioning. Moreover, most mixed-phase cloud studies have focused on the arguably simpler single-layer stratiform cloud structure, while relatively little focus has been given to multi-layer cloud systems. One of the major challenges with Arctic clouds, single or multi-layer, is the formation and properties of ice crystals. A great deal of research is now pointing to the potentially important role of secondary ice production (SIP) in shaping the phase composition of Arctic clouds, yet there are few modeling tools to study SIP and its role in cloud structure. Through a series of simulations, this manuscript examines different factors that impact the ice within multilayer clouds. Thus, thematically, the manuscript is timely and focused on a topic that is important for improving our understanding of Arctic clouds and their representation in models. The topic is appropriate for ACP.

The manuscript itself has a number of issues ("major revisions") that will need to be addressed before it is ready for publication. Many of the issues, listed below, are a matter of interpretation or description and should be straightforward to address. There are, however, two more significant issues that will require more work to address.

**The first is interpretation of results.** There are multiple examples outlined in the General Comments below where the authors speculate on why a given situation occurred. However, the speculation is not clearly labelled as such. Moreover, there is no need to speculate here because the results are from a model and the model should provide all information needed to clearly state why the given situation occurred within the model (which may or may not reflect nature). A good example is around lines 327-330 where there is an explanation given for the appearance of a second cloud layer. In my opinion there are physical inconsistencies in the explanation concerning evaporative cooling and mixing. It is possible that related mechanisms are in operation, but instead of speculating it is better to simply look at the tendency terms from the model to definitively state why the model formed the second cloud layer and then why it later went away. Temperature tendency terms (radiative, latent, mixing, etc.), water tendency terms (evaporation, condensation, etc.), and/or a buoyancy analysis would be very informative in this regard. Please

have a look at the many areas where "interpretation" is provided and then include supporting evidence from the model instead of just speculation.

→This comment is answered within the comment section for each instance.

The second significant concern is related to the model itself. Is it possible to evaluate the model's sensitivity to the specified ice properties and SIP mechanisms when the model representation of the liquid water is so far from reality? In general, there appears to be little sensitivity to the ice processes, and this lack of sensitivity might not be realistic. Based on many past model studies of Arctic clouds, there should typically be sensitivity to the specification of ice processes. For example, numerous papers and model intercomparisons have shown how increasing the ice nucleating particle concentration in the model leads to more ice formation (number and mass) and eventually full glaciation of the cloud. While different models have different thresholds for glaciation based on their own specific set of parameterizations, this basic behavior seems to be consistent across most models. Why is there so little sensitivity in this model? It could be that 1) the simulated cases are just so warm that there is not much that can be done to promote ice formation, or 2) the general model set up (the parameterizations and how they are implemented) is not fit for the purpose of simulating these mixed-phase clouds. There are other, contemporary model studies (not yet published) of very warm mixed-phase clouds that are having difficulty simulating ice formation, so #1 could possibly be true. But it is also important to ensure that this model can represent the basic processes that are known to occur in these Arctic clouds. The model should be run on a colder mixed-phase cloud case, like one of the classics from MPACE or ISDAC or a MOSAiC case from earlier in the year. At these colder temperatures, is the partitioning of phase better? Is there sensitivity to INP concentration? If the model cannot represent this arguably easier situation at colder temperatures, then there is clearly a problem with the model that would inhibit it from successfully assessing the sensitivity to various ice processes like SIP. If the model is able to represent reasonable sensitivities to INPs or SIP at the colder temperatures, then the challenges experienced for these warm cases might simply be due to the fact that the model's specific parameterizations are themselves not suited for the warmest temperatures.

→ This is the reason we are looking into this case in such detail. The model in its original setup is not producing ice at warmer temperatures due to limitations in the immersion freezing parameterisation. The new parameterisation for immersion freezing is our solution but we find it is not enough to produce observed levels of ice mass concentrations. It is sensitive to differences in INP however the observed INP concentration is very low, and we argue that this is not enough to initiate further ice processes in the clouds.

The phase partitioning is better in the model for colder clouds as shown in the first case study of the 1$^{st}$ of September. However, we focus on the warmer clouds due to the large discrepancy between the model and observed levels of ice mass

concentrations. The points you bring up are valid, but we may argue that we are showing the sensitivity of the model by changing the immersion freezing parameterisation and scaling this up shows that the model is responding, the same is true for the SIP simulations. The lack of large impacts is mostly seen for very low immersion nuclei at warm temperatures as well as the small changes performed for cold immersion and deposition nuclei. We have altered the language throughout the manuscript to better highlight the differences that are seen within the simulations (see comments with Lines 148, 268, 279, and 308-310).

In general, the model has been extensively evaluated over Germany as well as the Arctic and can quite accurately model Arctic clouds. We have added a sentence to emphasize this: "Microphysical perturbation studies on Arctic clouds have also been evaluated using the ICON model and its predecessor COSMO, which included the same cloud microphysics scheme (Stevens et al. 2018, Possner et al. 2017, Loewe et al. 2017, Possner et al. 2024)."

My final summary point is related to the last comment above. If the first order goal is to examine model sensitivity to ice nucleation and SIP processes, it is probably best to do so in a temperature range where these processes are known to be active and effective. At temperatures near 0 C, all ice processes are greatly diminished and even the WBF is not very effective. Thus, the selected cases are not actually great conditions for understanding INP/SIP sensitivities. It would be much preferable to examine these processes at -6 to -20 C where ice is clearly more significant and where a variety of SIP processes are expected to operate.

→ The main goal of this paper is to evaluate the model's capability of accurately modelling multilayer clouds. The second goal is to explore microphysical sensitivities for multilayer clouds, so the case was not chosen to optimally look at primary and secondary ice processes, it was chosen as an interesting multilayer cloud case.

As shown in Figure 6, the cloud top temperatures for the upper and lower layers are -8°C and -1°C respectively for the case we are interested in INP and SIP perturbations on the 3$^{rd}$ of September. This temperature range is of great interest for both warm-activating INPs as well as SIP. As we show in the SIP section 5.4, there is a large response of SIP in both the upper and lower layers (Fig. 9f,g). However, understanding SIP was never the main goal of this paper. Instead, we wanted to show the many ways the model is misrepresenting these warm mixed-phase clouds and explore the options to improve their representation.

General comments

Title: This is a detailed title, but I'm not sure it clearly represents the paper. First it is not clear that any of the simulations arrived at a "realistic structure and composition", so it is hard to say what is required to produce those. Additionally, as noted in some of the comments below, "efficient primary" ice processes do not appear to be in operation in these simulations, in large part because apparently the only primary ice nucleation mechanism is rain freezing, which is inefficient in the model and should

be inefficient in these clouds. Lastly "…and secondary ice processes" is also not reflected in this paper, as really the paper only dealt with one SIP process and did not examine the (presumably tunable) efficiency parameters embedded in that SIP parameterization. Thus, I think it is best to come up with a more representative title.

→ I think there is a misunderstanding about the main work in the paper. We aimed to improve the immersion freezing parameterisation by constraining this to observed INP measurements. This is the efficient primary ice nucleation pathway that we discuss. Furthermore, we have included three SIP mechanisms. Breakup upon ice-ice collisions turned out to be the dominant process while the rime-splintering and droplet shattering are present but have smaller impacts. We clearly show the impact of SIP and the need to include at least breakup upon ice-ice collision for a more accurate representation of the clouds.

We never explicitly state that we optimally represent the observed clouds, but we isolate the requirements to achieve this. The title, to us, reflects the requirements, not necessarily that we achieved that perfectly. Through the two days discussed in detail, we show that the structure can be accurately captured (on the 1$^{st}$) when the thermodynamics in the model is accurately capturing the observations, while to capture the cloud phase we require both efficient primary ice production through the immersion freezing as well as SIP (3$^{rd}$).

However, we have explored other titles and suggest:

"Sensitivities of simulated mixed-phase Arctic multilayer clouds to primary and secondary ice processes"

Line 47: There are "at least six" SIP mechanisms. Recent work has suggested more.

→ Added "at least"

Line 56: "shown" is a bit of a stretch here. At best, some studies have inferred that these processes might play a role, but little about SIP has been definitively shown in natural clouds based on observations.

→ Pasquier et al. 2022, referenced with this statement, shows nicely the presence of SIP in their observational study in the Arctic, thus the word "shown" is used.

Line 66: It is better to say "near the north slope of Alaska" as the flights themselves were often over the adjacent ocean.

→ Okay, changed.

Line 66-67: It seems that a definition for cloud needs to be given somewhere. Individual layers within a cloud sounds like it could be two clouds or it could be one, depending on the definition. Since this whole paper is about multi-layer clouds, it is important to give a clear definition for what is meant.

→ Line 59 gives our definition of MLCs.

 Line 91:  "moored to an ice floe"

→ Ok.

Line 95:  LWP is "retrieved" not "recorded"

→ Ok.

Line 96-99:  There are additional uncertainties for this type of cloud product specifically related to the cloud type classification, and unfortunately these are unquantified.  For example, while a given IWC retrieval might have a quoted uncertainty of 40%, that is when the retrieval is applied to the appropriate cloud. But if it is applied to the wrong type of cloud the uncertainty can be much higher. Cloud type classification is the challenge here and Cloudnet has some challenges in that regard.  If nothing more, it is worth mentioning that there are other uncertainties associated with the full way in which the cloud retrievals are applied.

→ We believe the errors associated with the retrieval are substantially covered in the papers covering the algorithms. However, we will acknowledge the difficulty in obtaining reliable classifications by adding this sentence:

"Cloud classification is a challenging topic. During the days investigated here, radar and lidar products were available, making the classification fairly confident and reducing the errors associated with the method."

Figure 1:  The caption discusses degrees C while the axis label is in K.  It would be best to have a consistent temperature unit used throughout the paper.

→ As the new immersion freezing parameterisation is given in Kelvin we would like to keep the x-axis in Kelvin but we have now added a second x-axis showing Celsius (see Fig. i below)

[Figure]

Fig. i: Updates to Fig. 1 in the original manuscript with an added second x-axis in Celsius and the INP constraint of 1 per L (pr_1L$^{-1}$ ) added (as suggested by Reviewer #1).

Line 122-125:  It would be very useful to know the vertical resolution in the boundary layer and/or at cloud level. I understand that the resolution changes in the vertical, but some information is needed on how well this model set up is able to resolve the appropriate cloud structures.

→ Sorry about forgetting this, we add this to the model setup section:
"This translates into a vertical grid spacing at the lower cloud top (~600 m) of about 55 m for the 6 km and the 1.6 km simulations. The 400 m and 100 m simulations have 39 m, and 32 m vertical grid spacing respectively."

Line 135-136: It is not clear what this statement means.  Typically, the "spin up" is to spin up the turbulence while the thermodynamic state is largely advected into the domain based on the model forcing. Certainly, there is also interaction between the turbulence and thermodynamic state.  Please clarify.

→ Sorry for the confusion, the statement is phrased a bit awkwardly. By this, we mean that if we let the model spin up for longer (before analysing the output) we lose the accurate thermodynamic state from the initialisation. I.e. the model starts deviating from the observed state. The new sentence reads;
"This relatively short spin-up time was deemed necessary to not substantially deviate from the observed thermodynamic state."

Line 141: We have gotten to the end of the description but so far there is no documentation of the spatial scale of the different model domains. This spatial scale information (similar to the vertical resolution information) is needed to understand how well the cloud systems are resolved within the domain. Without knowing this information, it is difficult for me to comment on the appropriateness of the applied domains and resolutions.

→ We apologise for the lack of information. The extent of the domains is now given in the caption of Figure 3 where the large domain extents are given in degrees and the small ones are given with the radius in km. It reads: "The 6km domain spans 60N-90N and the 1.6 km domain 85N-90N. The 400 m domain has a radius of 112 km and is centred on the ship location. The 100 m domains also follow the ship location with a 33 km radius."

Line 148: As stated here, CCN activation is based on vertical velocity. Yet, one of the conclusions of this paper is that the horizontal resolution doesn't matter much. It is not clear how this can be true unless the CCN is an insensitive parameter in this model set up. At 1.6 km resolution the individual eddies (i.e., updrafts) in these clouds are not well resolved, such that the grid-scale vertical motions are likely much smaller than the actual vertical motions that occur at the (smaller) cloud scales. Thus, the CCN activation is likely less and more homogeneous across the cloud

compared to the spatially inhomogeneous way that CCN are activated in natural Arctic stratiform clouds. It seems that a discussion of this point is quite relevant somewhere in the paper, especially its implication on the apparent insensitivity of the simulated clouds to the CCN perturbations. Additionally, it would be very informative to show vertical velocity results to provide insight into how well the model resolves cloud-scale processes as a function of resolution. There is literature (including some of the papers in the references section) that can provide insight into the expected magnitude of vertical air motions in these clouds.

→ We apologise for the unclear language. The clouds do change with a finer grid spacing and we have now altered the language throughout the manuscript to better represent this. In general, we were expecting larger impacts and thus the language has more reflected this lack of large impacts.

We have now added more figures to the appendix (see Fig. ii below) to better show the impacts of the simulations and discuss the changes in the resolution more in detail. Time-averaged (06-12UTC) cloud droplet number concentrations are shown and show large impacts due to changing the horizontal grid spacing (approximately 4-time (15-time) increase from the $pr\_1L^{-1}$ simulation for the 400m (100m) simulation respectively while at the same time, the cloud thickness decreases).

[Figure]

Fig. ii: (a) Cloud droplet number concentrations, (b) cloud water mass content, (c) cloud ice number concentration, (d) cloud ice mass concentration and, first and second saturation adjustments, effectively the cloud droplet condensation routine, called to adjust the excess or deficit of vapour. Panel (d) only shows the first four simulations as listed in the legend. All were calculated for the 3rd of September as a temporal and spatial mean between 06-12 UTC. Added as Fig.A2 in the new manuscript.

However, this is not the variable of interest for us as we are mostly concerned with better representing the cloud ice mass concentration which has a larger discrepancy compared to observations. When considering cloud ice mass concentration, the finer

grid spacing does improve from the pr_1L$^{-1}$ simulation by a factor of 4. However, this is still a small impact compared to the scalings required to reach observed levels of ice and we argue that the computational cost required to run sensitivity studies on this high resolution is not justified.

→ Histograms of vertical velocity are provided here in Fig. iii

[Figure]

Fig. iii: Histograms of vertical velocity from four simulations with different grid spacings. All grid points and heights within the evaluation region are plotted (same region as contours). The bin size is adjusted to the size, N, of the dataset by 2*N^(1/3).

→ Capturing how real clouds activate CCN with a model requires very fine resolution to resolve updrafts combined with prognostic aerosols capable of cloud formation. We believe a discussion on how CCN activates in natural clouds versus models feels out of the scope of this paper but we have added a statement to the discussion:

"With a prescribed CCN concentration and coarse horizontal grid spacing, capturing realistic cloud droplet activation is challenging. Prognostic aerosols using a dynamic aerosol model such as ICON-ART (Aerosol and Reactive Trace gases module) would improve the representation of the local CCN concentrations and provide a more realistic cloud droplet activation. Furthermore, accurate representation of updrafts ensures a more realistic simulation of cloud droplet activation, ultimately enhancing the overall predictability of cloud properties."

Line 153-154:  This sentence is repetitive with the following sentences and can be removed.

Ok.

Line 161-162:  Is there some justification for why "rain freeze" is the only primary nucleation mechanism for T > -12 C?  Most (all?) of the pertinent clouds in these simulations are within this temperature regime, such that immersion freezing, and other nucleation mechanisms are not important at all. This would then require rain to form before ice could start forming. "Rain" is not common in these cloud as there is simply not the moisture and dynamics to form rain drops.  There can be supercooled drizzle at these temperatures, so is that included in the "rain freeze" mechanism?  If so, then this point should be discussed more clearly. If not, then it is not surprising

that the lowest level cloud is typically almost entirely comprised of liquid water. Ice in that layer would only start to form due to what should be rare formation of rain or seeding from above (which could all then be multiplied by SIP).

→ Rain freeze was the only primary ice production mechanism in the original version of the model, this corresponds to the 1.6km simulation. We have clarified this statement by referring to Sect. 3.2 where the temperature threshold for immersion freezing is altered to better represent these warm clouds, we also clarify that rain includes drizzle.

Sentences have been altered to better describe this: "Rain freezing, which also includes the freezing of drizzle drops, is counted as a source of primary ice as rain droplets implicitly contain many INPs. "

and
"Rain freeze is the only ice production mechanism above -12°C currently implemented in the model, Sect. 3.2 introduces the changes performed on the immersion freezing parameterisation to improve on this representation." We have also added further emphasising sentences to ensure this is clear in Section 3.2 as well as in the result section when mentioning the INP scaling.

If the model says that there is a lot of rain forming in these clouds, then it is probably not properly representing natural clouds and the issue should be better understood.

→ Rain doesn't necessarily mean large droplets falling to the surface, but the formation of larger droplets inside the cloud due to collision-coalescence.

Finally, there is discussion (i.e, Line 193-196) about scaling all of the ice nucleation modes by a factor of 0.05. But when are deposition and immersion freezing active?

→ This is specified in Section 3.1.2.

The clouds simulated in the case studies are all warmer than the cut off thresholds for these two nucleation mechanisms. Then in lines 197-198 there is a statement about adding INPs at high temperatures (up to -7C), but it is not clear if this is an adjustment to immersion freezing so that it can occur up to -7 instead of only -12C. That point should be clarified.

→ We have added some clarifying sentences, thank you for pointing this out.

Even if immersion is adjusted to be active up to -7C, it still is not active in a lot of the clouds that are simulated.

→ Yes, some clouds (including the observed ones) are at temperatures above what is possible through primary ice production. This is why an investigation into the seeding and SIP is warranted as these are the only mechanisms possible to allow for ice at these temperatures.

Line 164-166 (and Line 440-441): "is known" is too strong here. Perhaps "is hypothesized." The community simply does not understand SIP well enough right

now to know what mechanisms are in action under what conditions, and what their net impact is on the ice properties.

→We can understand your wariness of SIP as these are mechanisms that are not fully understood. We hope our discussion section provides the critical view we share with you in regard to this.

To better reflect this view we change " is known" and  "are commonly thought" to "hypothesised" in Line 164-166  and Line 440-441.

There is a "gap" between measured INP concentrations and measured ice crystal number concentrations, and people speculate that SIP might fill this gap.  The "gap" between observations and models is another thing altogether. Surely it is possible to build and tune SIP parameterizations to fill any gaps that are present, but this does not confirm that SIP processes are actually the reason for the gap.

→ We apologise for the ambiguous language. We here speak about the discrepancy between modelled and observed ice crystal number concentrations that are commonly seen in the Arctic. Line 165 is now changed to "SIP effectively increases ice number concentrations and is hypothesised to "fill the gap" between observed INPs and measured ice crystal number concentrations as well as between measured and modelled ice crystal number concentrations."

Line 171-172: Same as my above comment. There are a number of statements throughout this paper that tend to push the conclusions about SIP beyond what can really be concluded. In this sentence "has been shown to have a considerable impact" is a challenge because these are model studies. In a model a given SIP parameterization can be tuned such that it has an impact on the modelled clouds, but that does not mean that those same processes are important in natural clouds. Some of the language here should be tempered to more closely reflect the state of understanding based on observational and laboratory studies, which is not definitive at this point.

→ We agree with you that SIP is not a process we fully understand, and we lack well-constrained parameterisations which further introduces uncertainties. We have tried to change the language to better reflect this uncertainty. This line now reads: "This process has been found to be weak when acting alone but in combination with ice multiplication from breakup upon ice-ice collisions, it has been shown to have a considerable impact in simulations of Arctic clouds" instead of "it has been shown to have a considerable impact on Arctic clouds".

Figure 5:  It is hard to read the contour labels, please replot with larger font.

→ We have increased the size of the star as well as the contour labels, we hope this is large enough to read. An updated figure can be seen below (Fig. iv) together with the trajectories (following suggestions from Reviewer #1).

[Figure]

Fig. iv: (a) Synoptic situation similar to Fig.3 in the original manuscript and (b) back trajectories similar to Fig. B1 in the original manuscript.

Line 246: I'm not sure I've heard "hydrometeor content" before. How about just "water content" as in the labels?

Ok.

Line 263-264: What does this mean? Is there data assimilation at other times?

→ New information regarding the data assimilation procedure has been obtained. We will update the manuscript accordingly. Please see the section at the bottom of this document.

Line 264-265: I'm not sure that the strength of the inversion is what sustains the cloud. It is likely more so the other way around. The inversion is not present because the cloud is not there to radiatively cool and drive vertical mixing. I also do not see the justification for the next statement about excessive vertical mixing being the culprit. Please provide further clarification / justification.

→We can agree that it is not always clear if the cloud is driving the inversion or the other way around. To avoid miscommunication, we have decided to remove this statement.

→In regard to excessive mixing, few studies exist on this, and we commonly blame excessive mixing without further proof, however, one reference has evaluated this, and we instead refer to this one. The sentence now reads; "ICON is found to place inversions too high in comparison with observations, a similar error has been found to be due to excessive vertical mixing in the European Centre for Medium-Range Weather Forecasts (ECMWF) Integrated Forecast System (IFS) model (Sandu et al. 2013)"

Line 268-269: From Appendix A it is not possible to determine if the simulations are improved. On the 1st, the figure set up and chosen contours do not allow one to see the potential impact on resolving cloud-scale motions and variability, which would be

expected with higher resolution. On the 3rd, it does look like the higher resolution is starting to resolve some pulses of ice formation. I believe that starting to resolve some of these structures is actually a step in the right direction.

→ As we state in the discussion, higher resolution may definitely be helpful for certain analyses. We decided to not spend the computing resources on more high-resolution simulations as they did not improve enough compared to the cheaper simulations. We have now clarified our language:

"Further increases in resolution beyond 1.6km have large impacts on the cloud droplet number concentration (Fig.A2a) due to higher vertical velocities resolved with smaller grid spacing. Cloud ice is less affected but shows an increase with the 100~ simulation. This is shown in Appendix A. Due to the computational cost of running the high-resolution simulations the improvement (on especially cloud ice) is found to be too small to further study microphysical perturbations with this setup."

A new plot of the 100m contours is provided as it was noticed this simulation was plotted at a higher temporal frequency than the other simulations. For better comparability, this has now been amended.

We have now added vertical profiles of cloud mass and number concentration (of both liquid and ice) to the appendix (see Fig. ii and Fig. v below) and added a discussion on the increase in droplet number concentration with a smaller grid spacing.

On the 3rd:

"The time-averaged (06-12UTC, Fig.A2b) maximum liquid water content in the lower layer barely increases between the runs. In terms of droplet number concentrations larger impacts are seen with a 4-time (15-time) increase from the $pr\_1L^{-1}$ simulation for the 400m (100m) simulation respectively (Fig.A2a). This shows the impact of decreased horizontal grid spacing on the CCN activation. With a smaller grid spacing an increase in vertical velocity is found which increases the CCN activation. An increase in cloud ice by a factor of 4 can be seen for the 100m simulation (Fig.A2d), but this is deemed to be not enough impact to justify the large computational cost of performing microphysical sensitivity tests at such high resolution."

On the 1st:

"The time-averaged (10-12UTC) vertical profiles are shown in Fig.A4. An impact on cloud liquid is noted with a 3-time increase in droplet number concentration for the 100m simulation compared to the $pr\_1L^{-1}$ simulation. Meanwhile, the liquid water content only increases by a factor of 1.1. Cloud ice mass content in the upper layer increases by a factor of 2 while the ice number concentration increases by 5. In contrast, the lower layer does not see an increase in cloud ice with a smaller

[Figure]

Fig.v: (a) Cloud droplet number concentrations, (b) cloud water mass content, (c) cloud ice mass content, and (d) cloud ice number concentration. All were calculated for the 1st of September as a temporal and spatial mean between 10-12 UTC. Added as Fig.A4 in the new manuscript.

Line 278-279: Based on some of the comments above, it is not surprising that there is no large impact on the lowest cloud. Since only rain freezing is possible at the given temperature of this cloud, and rain formation should be very rare in these clouds, it doesn't really matter how many INPs are present.

→ The upper cloud discussed here spans most of the troposphere and thus most temperatures for heterogeneous freezing. A sentence is added clarifying which layer is discussed. Furthermore, we add a longer discussion on the changes seen in ice mass concentration to highlight that we do see differences with a change in primary ice production:

"The upper layer shows small changes due to the scaling of the heterogeneous freezing/nucleation parameterisations. A reduction in INPs (at cold temperatures above 4km, see Fig.8e) induces small reductions in ice mass concentration in the upper levels of the upper cloud (above 4km). This amounts to a time-averaged (10-12UTC) reduction by a factor of 0.5 in the ice mass concentration. In the lower levels of the upper cloud (at warmer temperatures), an increase is noted. The time-averaged difference amounts to an increase by a factor of 1.3. The lower layer is unaffected by the changes due to its high temperature (Fig.6a,b)."

Line 284-286: This interpretation is not convincing for a couple of reasons. First, just based on Clausius-Clapeyron, sublimating ice will not provide enough moisture to then lead to liquid water saturation without a significant cooling of the air parcel. That cooling would have to come from vertical lifting of the parcel. But how would the parcel be lifted, ie. what provides the buoyancy? The text suggests that this is due to latent heating (i.e., condensational heating). But the source of moisture was from sublimation (cooling), which would cause the parcel to sink not rise. I don't think it is possible to have both sublimation and buoyancy generated from condensational heating at the same time. Perhaps I'm missing something that needs to be clarified?

In general, the model should provide the information that is needed to understand the thermodynamic balances at play as a function of height and to clearly distinguish why the model produced liquid water. Generally, there will need to be some convergence of moisture at that height, likely supported by advection (as suggested by the soundings), and the ice deposition rate must be small enough that some liquid water can form. Once that liquid water forms, the typical mixed-phase processes will kick in (radiation-turbulence-microphysics feedback) to allow the layer to persist for some time in the face of the low ice crystal concentration. To help understand this situation, it might be useful to do a simulation where ice is turned off altogether. I suspect that liquid water will form at that height supported by moisture advection. Then, as ice is turned on, and turned up, eventually the liquid cloud cannot sustain itself (as is shown by some of the simulations).

→We agree that we have not provided enough evidence for our line of thought. We have removed the discussion regarding the liquid layer appearance as it also does not contribute much to the storyline.

Line 287-288: Where is the evidence of this seeding? It looks like there is a full gap between layers without any falling ice in between.

→ Fig. 7 shows contour lines of snow and graupel entering the lower layer between 10-12UTC. A reference to this plot is added after Line 289 to clarify.

Line 289-290: The fact that the cloud layers "seem quite impervious to perturbations" is concerning. There is a lot of literature on modeling studies that show clear sensitivities to ice. It should be possible to turn up the INP concentration high enough to achieve glaciation in the lowest cloud. So why does this model not show much sensitivity? It could be that 1) the cases are just so warm that there is not much that can be done to promote ice formation, or 2) the general model set up (the parameterizations and how they are implemented) is not fit for the purpose of simulating these clouds. What happens if this model runs a colder mixed-phase cloud case (like one of the classics from MPACE or ISDAC or a MOSAiC case from earlier in the year)? Is there sensitivity to INP concentration?

→ This is answered above under "The second significant concern". To be added here, yes, for this cloud, option 1) is true. To achieve glaciation of the lower layer on the 1st we require very efficient seeding as there is no primary ice production in the lower layer due to its high temperature (cloud top temperature about -3°C). We achieve some seeding (pr_1L$^{-1}$ simulation) but this weak seeding is not enough to impact the lower layer due to the inefficient glaciation at this temperature. The seeding impact will be the focus of future publications so to limit the extent of this paper we decided to not perform further scalings of sedimentation velocity and INPs to perturb the lower layer through seeding.

Line 308-310: WBF can only glaciate the cloud if there are enough ice particles. To better understand this point, what are the actual ice number concentrations?

→Thank you for pointing this out, we have indeed not shown any ice number concentrations. Some of the time-averaged (06-12UTC) ice number concentrations were previously shown in Fig. B6. This plot, shown below (Fig. vi), is updated in the manuscript and moved to the main text (new Fig. 11). For INPx1E4 and INPx1E6 we are looking at maximum ice number concentration values in the upper layer of 0.22 per L and 6.7 per L respectively, compared to 0.00060 per L for the INP simulation. We have now added these values to the text with references that also do not see glaciation at these ice number concentrations.

"With this high INP concentration, the ice number concentration increases as well. Time-averaged (06-12UTC) maximum ice number concentrations in the upper cloud layer reach 0.22 per L and 6.7 per L for the INPx1E4 and INPx1E6 simulations respectively (Fig 11e). This is compared to 0.0006 per L for the INP simulation, an increase of two and four orders of magnitude respectively. However, irrespective of this substantial increase in ice number concentration, these values do not necessarily lead to the glaciation of the cloud (Stevens et al. 2018, Solomon et al. 2018) and this phenomenon is not observed in these simulations either."

And new Fig.11:

[Figure]

Fig. vi: Updates to Fig.B6. Two new runs have been added; INP+CCN+SIP scaled and 1.6km+SIP. Vertical profiles of other runs have been added and the ice enhancement factor is shown as a vertical profile (f).

We further add a comment on the ice number concentrations reached for the SIP simulations within Section 5.4:

"Time-averaged (06-12UTC) maximum ice number concentrations in the lower cloud layer reach 484 per L, 396 per L and 156 per L for the 1.6km+SIP, SIP and $pr_{1L}^{-1}$ + SIP respectively (vertical profiles are shown in Fig.11e). The large increase in ice number concentration drives the increase in vapour deposition through the WBF process (Fig.B2c,f) resulting in the glaciation of the lower layer."

Is it the ice number concentration that increased by two orders of magnitude or only the INP concentration?

→We apologise for the ambiguous language. Here we mention the ice mass concentration; this has now been remedied in the paper. The INP concentration has been increased by 1E4 and 1E6 respectively for the simulations discussed here.

In these simulations it seems apparent that primary nucleation is a limiting factor such that there are not enough ice particles to make WBF much of a factor; if the available INPs were to nucleate into ice crystals there should be plenty of ice for the WBF (although WBF is also limited at these warm temperatures). Generally, these results seem to suggest that the number of INPs is not the problem with this model set up, but it is rather the parameterizations for ice crystal nucleation.

→ We have tried to improve the language throughout the text to better show the impacts of changing the primary nucleation. We do see impacts as discussed further above and we hope through these new additions this is shown more clearly.

Line 312: Here the CCN activation rate perturbation is introduced. It would be useful here or earlier when the simulations are introduced to more clearly outline why the different simulations are performed. In this case, for example, why is a perturbation of the CCN implemented? Is there some hypothesis that the properties of the liquid drop size number/distribution are important in the ice processes? Please explain so that the reader understands the logic behind the different simulations.

→ Line 209 states the CCN activation scaling. This scaling was decided as the model produces too much liquid water.

We add a clarifying sentence "To investigate the impacts of the cloud droplet activation on the too-thick modelled liquid layer, a sensitivity study with a decreased cloud droplet activation rate (pr_$1L^{-1}$ + CCN) is performed."

Line 319: It looks to me that the LWP is 3-4 times too high and not 5 times too high.

→ Yes, you are right. 4.2 to be exact, we will change to "4 times".

Line 327-330: The explanation for the appearance of this second layer seems to be speculative and it is not clear why evaporative cooling would lead to mixing. Additionally, there is speculation in the following sentences about the upper layer then radiatively shielding the lower layer so that it "dissipates." It is possible that these mechanisms are in operation, but instead of speculation it would be best to look at the tendency terms from the model to definitively state why the model creates another layer and then why that layer goes away. For example, temperature tendency terms (radiative, latent) or water tendency terms (evaporation, condensation) will be informative in this regard.

→As we have not obtained these tendency terms at the time of the simulation we will simply remove the speculation in regards to the formation of a third layer. A sentence

in regards to this now states: "The cloud structure changes during the time window 07-09UTC where the presence of a third layer is evident."

Line 333: This statement about the upper layer causing the lower to dissipate is likely true and has been described by a few papers both observationally and within models. However, the description of this process has revealed the point that the authors tend to describe what is playing out in this Eulerian perspective as if it were a Lagrangian perspective. While the additional cloud layer only lasts for a couple of hours within the stationary vertical column, this perspective does not represent how long the cloud itself actually lasts. This could simply be a second cloud deck that advects into and then out of the vertical column of interest. When it leaves the vertical column, this does not necessarily mean that it dissipates as it is also advecting at the time. The authors should consider the difference between Eulerian and Lagragian perspectives when describing the interactions and transitions, both here and throughout the manuscript.

→Yes, you have a valid point. We have played around with an upstream perspective as well but that has not been used for this analysis. We have decided to remove the discussion about the generation of the third layer due to the lack of advective tendency outputs from the model.

Line 336: Which increase in condensation?

→ The increased condensation (due to the saturation adjustment) in the "third layer" was shown in Fig. B4e in the original submission at a height of about 1200m. This plot has now been updated and the condensation rates are shown in Fig. A2 in the revised manuscript.

We clarify: " With the presence of more cloud liquid, due to the presence of a third cloud layer (Fig.9e), condensation increases due to the saturation adjustment within this layer (Fig.A2e) and the resulting impact of the CCN scaling is small."

Line 355: "ice concentration": Does this mean ice number or mass concentration? Please be clear.

→ We have now edited all instances where this is ambiguous, thank you for pointing this out!

Line 355-356: This statement could possibly be true under certain circumstances, but it should be shown with model results. This process would totally depend on the ice crystal number concentration, which is not given. In general, even for enhanced ice cases, the number concentration of ice crystals is still likely very low. When one considers the size of a crystal, and the number of ice crystals and liquid droplets per volume, there is typically a large physical spacing between ice crystals with many liquid droplets (i.e., lots of surface area for evaporation) in the vicinity of each ice crystal, providing ample vapor supply to grow the ice.

→We may remove this part of the sentence as it does not provide much more information for the next part.

Line 381-383: this collisional breakup is a positive feedback, where more particles makes more collisions, which makes more particles. Thus, it is indeed an attractive mechanism to multiply ice concentration. This also means that the decisions made regarding the parameterized efficiencies (collision efficiency, breakup efficiency, number of resulting particles per breakup event, etc) are very important and some might be highly sensitive. This point should probably be discussed.

→We briefly discuss this point in the discussion section where we deemed it not worth the computing resources to arbitrarily tune these parameters to our setup. Instead, we provide a simulation using the scaled breakup upon ice-ice collision parameterisation, which uses the diameter of the colliding particles for a more physical constraint of the breakup process.

Line 384-391: Given that seeding plays only a small role and does not appear to be the culprit for glaciation towards the end of the case, why does this glaciation occur?

→Thank you for bringing this point up. We have investigated and it seems the SIP in the lower layer (which is the more obvious glaciation as the upper layer dissipates in all runs) is driven by a small change in temperature in the lower layer at 10UTC-11UTC, please see the plot below (Fig.vii). Even a small change in temperature substantially changes the number of fragments generated within the breakup parameterisation.

[Figure]

Fig. vii: Time-height plot of temperature for the SIP simulation. The explosive tendency in the breakup upon ice-ice collision mechanism is governed by the temperature drop starting at 10UTC in the lower layer. Contours show liquid water

content plus ice water content at a specified contour at 10^-5 g/m$^3$ to outline the cloud.

We update the manuscript:

"The breakup upon ice-ice collision parameterisation is highly sensitive to temperature fluctuations and the sudden onset of glaciation through ice particles generated by SIP is due to a drop in temperature of the lower layer."

The SIP processes are presumably occurring for many hours prior to the point of full glaciation. What is special about that transition? Line 398-399 further suggests that there is a time dependence to the SIP impact. First, it is difficult to interpret time-dependence in this Eulerian perspective. Second, if there is indeed time dependence, what is the mechanism for this?

→We talk about time dependence from this Eulerian perspective with regard to how long the simulation has been running but we agree that it is ambiguous. We have now removed this statement.

Line 393-395: I think this sentence states that there are no sinks of ice and that the ice that occurs at the analysis point was initialized when the model was initialized and advected all the way to the analysis point (i.e., it is the same ice). However, I do not believe this is the case unless there is something strange with how the model represents fall speed. Ice crystals typically fall at about 1 m/s, so they would fall more than 3 km in 1 hour. Thus, there is a sink of ice crystals from any given layer of the atmosphere due to fall speed. Moreover, the residence time of ice in the atmosphere relative to advection from the domain boundary is something that can be determined directly from the model.

→We apologise for the unclear statement. We simply mean that some of the cloud ice within the domain is not produced during the simulation due to the initialisation and the updating boundaries. However, there is a lack of strong sinks such as large rates of snow formation. Regarding sedimentation, we have calculated the sedimentation velocities of the ice crystals through mean ice crystal mass and found these to be low. Through a mass-diameter power law, ice crystal sizes vary around 60-80 micrometres (close to the cloud top on the 3rd of September) in diameter which gives a sedimentation velocity of about 10 cm/s. This means it takes ice crystals found at the top of the upper layer on the 3rd about 6 hours to fall to the surface indicating that the ice seen at the ship location is indeed replenished through the simulation.

We update the line; "Here, as we are dealing with a real setup, some of the cloud ice within the domain is supplied from the continuous advection of cloud hydrometeors through the domain boundaries."

→ Unfortunately, no advective tendencies were collected during the simulation.

Line 433-435:  It is not clear how the extent of these clouds should matter. Please explain further. There are many past model studies that have been set up in a similar way and have shown sensitivity to CCN, so why not in this model?

→We realise our language does not reflect our results as well as we would have hoped. We were surprised by the lack of large impacts and thus this language has followed within the manuscript. Of course, we do see impacts and these have now been clarified throughout the manuscript as well as here:

"The reduction of CCN activation within the model impacts the structure of the clouds and the appearance of a third layer is evident. This reduces the impact of the reduction of CCN due to increased condensation within this layer and only a small change in LWP (reduction by a factor of 1.28) can be seen."

Line 437:  "…does not survive for more than two hours."  Given the Eulerian perspective you can only say that the cloud does not exist in the analysis column for more than two hours.  However, within the model you could track the cloud elements to see how long the cloud actually lasts along its Lagrangian trajectory, assuming the domain is sufficiently large.

→Yes, you are right, this statement is misleading, we have removed it.

Line 455-456: Indeed, the theta profile from the case suggest that there is not strong coupling with the near surface, such that the surface aerosol measurements are likely not representative of cloud level. In fact, the INP concentrations are likely larger aloft (based on some other work from MOSAiC and elsewhere).

→Yes, thank you for the comment. As stated in the manuscript we agree with your reasoning.

Line 467:  It is more useful to know the actual vertical resolution at cloud level than the number of layers in the simulation.

→This is now added to the model setup section. The sentence now reads: "The 6km and 1.6km runs are kept to 90 levels while the 400m (100m) run is increased to 150 (200) vertical levels. This translates into a vertical grid spacing at the lower cloud top (~ 600m) of about 55m for the 6km and the 1.6km. The 400m and 100m simulations have 39m, and 32m vertical grid spacing respectively at a similar height."

Figure A2: This comparison suggests that there is way too much ice/snow formed in the upper cloud. What does this say about other problems with ice nucleation and growth? Also, it is not possible from these plots to evaluate the impact of resolution on the lowest clouds, which should be the ones that are most impacted by resolution. It looks like no ice is forming in them, leaving way too much liquid.

→ This is discussed within section 5.2, where we tune down the heterogeneous freezing mechanisms at colder temperatures. The lower layers are too warm for heterogeneous freezing to occur, they can only be seeded from above which is

something that we see in one of the simulations on the 1st but it is weak and does not impact the lower layer.

We have added vertical profiles in the appendix that may improve the analysis of the lower layer.

Figure B1: What is the gray shading? That should be added to the caption.

→ Thank you for noticing. We have now added a line explaining the sea ice concentration extent in grey contours.

Figure B5: What is the "heterogeneous freezing" and why is it at the bottom of the upper clouds? I assume this is the rain freeze mechanism and thus there is freezing at the bottom where there is the most rain? How are the particles at the top of the cloud formed? All of the particles will, on average, fall so there needs to be a particle source at the top, not just the base.

→ Heterogeneous freezing is mostly immersion freezing but also includes small additions from rain freeze. It is present all through the cloud layer, however, note the x-axis, comparatively to vapour deposition the rate of heterogeneous freezing is very low. To make this more clear, we have edited the plot such that vapour deposition is the last to be stacked. The new figure is shown below in Fig. viii.

[Figure]

Fig. viii: Process rates as shown in Fig. B5 in the original manuscript, here with a change of plotting order to highlight the heterogeneous freezing and the new runs added to the analysis (as suggested by Reviewer #1) (INP+CCN+SIP scaled and 1.6km+SIP).

Line 567-568: This citation is incomplete. Should include: Atmospheric Research, 51, 45-75.
Line 576-577: This citation is incomplete. Should include: Atmos. Phys. Chem., 12, 9817-9854.
Line 639-640: This citation is incomplete. Should include: J. Atmos. Sci., 62, 1665-1677.
Line 646-647: This citation is incomplete. Should include: Nat. Geosci., 5, 11-17.
Line 671-672: This citation is incomplete. Should include: J. Atmos. Sci.
Line 691: This citation is incomplete. Should include: J. Atmos. Sci., 63, 697-711.
Line 723-724: This citation is incomplete. Should include J. Climate
Line 736-737: This citation is incomplete. Should include J. Atmos. Sci.

→ Thank you for finding these discrepancies, all have now been edited.

Additional changes:

New information regarding the data assimilation strategy of MOSAiC radiosondes has been obtained. All radiosondes were assimilated from the campaign. Variables include temperature, wind profiles, and relative humidity. We are updating the manuscript accordingly.

---

## Author Response (AR2)

**Author's response to reviewer 2, revision 2**

We thank reviewer #2 for their detailed feedback on the revised manuscript. We have responded to each comment in detail below. A simple "Ok" means we have changed it, for larger changes, the sentence or figure in question has been included. Figures shown in this document are labelled using Roman numerals and referrals to the original figures are included when appropriate.

**Review 2**

After its first round of revisions, this manuscript is improved relative to the first version. There are fewer speculative statements, and some details have been described better. When pushed by both reviewers to add depth via additional simulations and analyses, the authors have instead argued for why these are not necessary. Fair enough; the authors can ultimately determine the scope of their study. At the same time, its impact on the community will be similarly scaled. The topic of ice processes and SIP in Arctic clouds is important, and the community does need more studies of this nature to help make the needed advances in understanding and modeling. Thus, perhaps this study is a baseline for very warm Arctic mixed phase and multi-layered clouds that can motivate the modeling community to dive further into the relevant processes. In this regard, it is useful to eventually see this manuscript published. There are, however, still some important details for the authors to address before the paper is ready for publication. All of the comments below can be addressed via text modifications, thus I would characterize these as minor revisions.

→ We thank you for your thorough comments. We will answer your points as they come up in the next section.

**The Main Challenge**

The main persisting challenge I have with this paper is encapsulated in the first paragraph of the discussion/conclusion section. The text outlines the two foci of the paper: 1) "to show that multilayer clouds can be accurately modelled…." and 2) "to evaluate the microphysical sensitivities…..while constraining the parameterisations to the observed ground-based measurements to better represent cloud phase." The first of these statements suggests that in this manuscript the multilayer clouds have been accurately modelled. This depends strongly on the definition of accurate. If it is defined as simulating clouds at approximately the right times and heights, then perhaps it can be argued that these simulations were indeed accurate. But I argue that this type of accuracy is simply due to the advective forcing of the model (i.e., where and when is moisture advected at the large-scale). In my mind the main point of the paper, as conveyed in the title, is rather about microphysics, and if we refer to Figure 10 we see that the clouds are not accurately simulated.

→We understand the ambiguity of the word "accurate". Our background to using this word even though the clouds are not perfectly captured comes from the fact that multilayer clouds have not been thoroughly evaluated in the high-Arctic with a realistic setup. We wanted to first make a point that they can to a certain degree be accurately captured with ICON to support further studies on these clouds.

We will add some distinctions to make the point that we do not perfectly capture the clouds clearer. We will split the focus into a thermodynamic and microphysics part. Thermodynamically, the 1st of September is accurately modelled while the 3rd was not, due to its initialisation as you stated. Microphysically, we do not capture the clouds very well. This is hopefully still encapsulated in the second point.

We change the line;

".. 1) to show that the thermodynamical structure of real cases of multilayer clouds can be (to a certain degree) accurately simulated…"

Arguably the most important aspect of these clouds is the super-cooled liquid water, which ultimately determines the impact of these clouds on the environment and defines the cloud life cycle. The broader Arctic mixed-phase modeling community also appears to agree that the liquid component is the most important, based on the existing modeling studies in the literature. These simulations do not get the liquid component of the clouds correct and do not show any simulations that significantly impact the liquid component, on average (e.g., Fig. 10). Thus, in my mind, these simulations cannot be considered "accurate." The second focus is on sensitivities in the microphysics. The paper has indeed examined sensitivities for the ice component, but interestingly, the simulations show no corresponding sensitivities for the liquid component. Thus, while in the most extreme sensitivity tests the observed ice component is indeed approximately reproduced by the model, there is no change in the poor representation of the liquid. So technically the cloud phase is "better represented," going from very bad to not quite as bad, but it is not well represented.

→ We thank you for bringing up the liquid water in this discussion. We did not focus on the liquid component of the clouds primarily due to the large discrepancy in cloud ice water content. The liquid water differs by a factor of 4 compared to more than two orders of magnitude difference in cloud ice. Furthermore, we find that reducing cloud liquid is difficult in a model based on saturation adjustment. Cloud liquid will always (as long as cloud droplets are present) form during supersaturated conditions. Thus, we do not focus on this but show the impact of reducing the CCN concentration, which does reduce the liquid water path by 20%. We also explain the reason this change is not more substantial (saturation adjustment and upstream changes to the clouds). On the sensitivities, yes cloud liquid does not drastically reduce but this cannot be expected as the cloud ice, even at the largest scaling of INP still only reaches 10 g/m^2. We do see some sensitivities to cloud liquid though, for example in Lines 371-372, we see a 77% reduction in cloud liquid due to an increase in, primarily, vapour deposition with the large INP scaling (1E6). The lack of glaciation may be surprising but the clouds are found at warm temperatures where glaciation is less likely (Line 318).

But you are right, it is in itself an interesting finding that the cloud liquid is so persistent and we add a further discussion to the discussion section regarding this (see further below).

Moreover, it is worth pondering if the sensitivity to ice processes would be different if there were less liquid water (like in the observations) or if there were more sensitivity of the liquid water to the ice (as is seen in many other model studies). So are these sensitivities to ice parameterizations representative?

→We do believe the sensitivities would change if cloud liquid was lower. However, we still believe these to be representative sensitivities as we do not have unreasonably high levels of cloud water. Future case studies of multilayer clouds with contrasting thermodynamic structures and within different large-scale flows would be beneficial to elucidate their sensitivities. We will, however, leave this for future work.

I still believe this paper has interesting results that contribute to the field, but it is important that the authors present the paper in the proper context. First, there needs to be some discussion of the liquid component of these clouds. Why does the liquid component show

such little sensitivity to the perturbations in the ice? Why does the model produce too much liquid to start with? And why is the model seemingly unable to produce sustained liquid water clouds that have much lower (more realistic) liquid water paths? One main goal for why we care about the ice is to properly simulate the phase partitioning in these clouds, so this topic must be discussed more.

→It is difficult to trace the origin of too much liquid. We believe the saturation adjustment is likely a culprit, together with misrepresentations in the initial and boundary forcings (1-moment ICON) which initialised and updated the boundaries with poorly constrained data. Liquid introduced into the domain may continue the cycle through the saturation adjustment while limiting the cloud droplet activation hardly influences cloud liquid water content given a pre-existing liquid cloud and the use of the saturation adjustment. We find that the saturation adjustment is not recommended for extreme maritime cloud cases (Kogan & Martin 1994), where the low CCN (~25cm^-3) gave rise to up to 40% error in condensed water vapour. However, follow-up work, about to be submitted (published in the thesis available here; DOI: 10.5445/IR/1000179667), where we look at a long-term simulation over the same area, has a lower prescribed CCN concentration and systematically under-predicts LWP compared to retrievals. This case might thus be an outlier.

Plotted below are the domain (85°N-90°N) and time (08UTC-12UTC) average process rates contributing (nucleation, 1st and 2nd saturation adjustment) and reducing (rain formation (accretion+autoconversion), riming by cloud droplets, and vapour deposition (onto ice, snow, and graupel) which within mixed-phase clouds shows the impacts of the WBF process) cloud liquid water content. Condensation (1st (left) and 2nd (right) saturation adjustment) is strong in this case. The rate for the 2nd saturation adjustment (right) is two orders larger than both the other contributing and reducing rates. While it also shows large negative rates (evaporation) it serves as a large source of cloud liquid within the cloud layers. Rain formation as well as mixed-phase processes are negligible in comparison and thus cloud liquid persists.

[Figure]

[Figure]

Fig I. Domain (85°N-90°N) and time (08UTC-12UTC) average process rates during the 3rd of September (1.6km simulations) contributing to (nucleation, 1st and 2nd saturation adjustment) and reducing (rain formation (accretion+autoconversion), riming by cloud droplets, and deposition (onto ice, snow, and graupel)) cloud liquid water content.

Future work should try to disentangle the complicated processes leading to this large impact from the saturation adjustment, including possible issues in CCN-limited regimes. Yet, we consider this out of the scope of this paper as the saturation adjustment in ICON has been thoroughly evaluated across the world including the Arctic (Schemann & Ebell 2020, Kretzschmar et al. 2020, Kiszler et al. 2023).

Considering the mixed-phase processes, an inefficient WBF process may also be blamed, this has been recently seen in ICON (Omanovic et al. 2024).

We add a discussion point on this subject in the discussion:

"The saturation adjustment (condensation) in the model ensures that during supersaturated conditions (with respect to water), condensation occurs, making a reduction in cloud liquid difficult to obtain. The initialisation of cloud liquid (using 1-moment ICON at a coarser resolution) enables the saturation adjustment to act throughout the simulation (as long as nucleation still supplies newly activated droplets and/or cloud liquid is not entirely removed). Overall, cloud liquid is persistent. While it responds to changes in CCN and large perturbations of INPs, we find it difficult to reduce the modelled liquid water content to observed levels. This can potentially indicate a lack of efficiency in the WBF process as seen by previous studies (Omanovic et al. 2024) or a possible issue using a saturation adjustment scheme in a low CCN environment (Kogan & Martin, 1993)."

Second, the authors need to provide the justification for why it is still relevant to examine the sensitivity to various ice processes in these cases where the liquid water is so far from accurate. Are these sensitivities generally representative and therefore relevant for thinner (more accurate) liquid water clouds? i.e., is it OK that the liquid component is not well simulated? I think the authors can make these points, and they need to make these points.

→As with all case studies, the applicability to other cases is limited. But we believe the sensitivities shown here bring up the important point that these clouds respond very similarly to single-layer clouds and this may be a transferable generalisation.
We add this to the discussion;

 "The sensitivities shown here highlight the point that these clouds respond very similarly to single-layer clouds."

The main purpose of the sensitivity study is to try to achieve the observed state. The sensitivities are introduced through the text as the next step to perturb the system into something observed rather than a study to test how the clouds react to sensitivities. Thus, we believe it's highly relevant to do the sensitivity tests introduced in the paper. Your point makes us realise we may not have made this clear enough and add this to the discussion as well partially in the introduction:

"The model in its original form, severely underestimates the cloud ice mass concentration (by two orders of magnitude) while the cloud liquid water content is over-predicted by a factor of four. We perturb the aerosol parameterisations in an effort to 1) obtain the observed state, and 2) understand sensitivities in the cloud response to aerosols."

Specific Comments:

Line 0: The title is much improved.

→ Thank you, we agree

Line 94: "ice floe" not "ice shelf"
→ Ok, changed.

Line 96: "….taken from level 3 files produced from Vaisala…."
→ Ok, changed.

Line 102-103: Thanks for mentioning the cloud classification is challenging. However, the newly included statement is misleading. Yes, having radar and lidar helps with cloud classification. Yet, in multilayered clouds (the focus of this study), the lidar signal is often attenuated by the lower cloud, rendering the phase classification in the upper clouds very unreliable. This is a distinct weakness of the CloudNet approach and adds significant uncertainty.

→Yes, the uncertainty is significant. We change the sentence to the one below to make this more clear.

"...During the days investigated here, both radar and lidar products were available for Cloudnet retrievals. In general, the highest classification confidence is given for clouds that are detected by both lidar and radar. Confidence decreases for the upper clouds in the MLC systems considered here as the lidar signal is attenuated by the lowermost cloud. For identifying MLCs, however, we utilise soundings and confirm the presence of clouds in saturated layers through lidar observations."

Line 103-104, 162: CCN and INP acronyms have already been introduced in the first section.
→ Thanks for noticing. Removed.

Line 106: "…..and from the US Atmospheric Radiation Measurement (ARM) facility…."
→ Ok, changed.

Line 108: Introduce the CSU acronym.

→ Thank you for noticing, changed.

Line 153-155: I understand the computational cost of high resolution. The manuscript mentions that a simulation was done at higher resolution, but with the baseline (default) approach to ice processes. According to the statement given here, this higher resolution did not have as big of an effect on the ice as the liquid. However, many of the other sensitivity simulations employ different details about the ice that might be sensitive to resolution. If the authors simply do not want to do further high-resolution simulations, that is acceptable, but a general statement like provided here that cloud ice is less effected by resolution seems to be inappropriate. It is possibly true for the one tested model set up, which itself already has many other problems, but unless the simulations are run, we do not know if it is true for some of the other ice processes that are tested and might produce more ice. Moreover, the fact that the high-resolution simulation gets some change in the liquid, when the other simulations conducted in this paper apparently do not, is very useful for addressing a major deficiency of the current paper – the fact that there is little impact on the poorly represented liquid phase by changing the ice processes.

→ Yes, we only tried one setup as it is very expensive to run and we found a lack of immediate improvement in the cloud phase partitioning. Of course, this does not show any sensitivities at finer resolutions, but simply that the baseline simulation at a fine grid spacing is similar to simulations at a coarser one.
We add for clarity:

"The cloud ice is less affected than the cloud liquid in this specific case."

We believe the induced differences in cloud liquid are quite well understood with a higher updraft velocity as more cloud droplet activation occurs. The reduction of cloud liquid is really where the challenge lies. We refer you to our answer above for this discussion.

Line 352-353: I believe this is an incorrect assumption. In the observations there is at least a 1-2 km gap between the upper and lower clouds. It takes time for ice to fall across this depth. The lower cloud layer is less than 1 km thick. Thus, the timing for seeding vs fallout from the lower cloud layer does not work out for this hypothesized mechanism for forming ice in the observed lower cloud. Moreover, why would there have been seeding ONLY upstream for this whole time period? There is no clear reason why that would happen upstream consistently over time but not at the observation site. Typically, if there were seeding upstream the whole system would advect in time and eventually be observed at the Eulerian observation point. It is much more likely that either: 1) The "observations," which are really retrievals, are wrong about the presence of ice in the lower cloud, or 2) there is a mechanism for forming ice in the lower cloud that does not involve seeding.

→The IWC retrieval (Hogan et al. 2006) in Cloudnet comes with high uncertainty. Furthermore, in this specific case, with temperatures close to zero we are close to the boundary of the valid retrieval range which adds additional uncertainty.

We will add a point on the uncertainty of the retrieval together with this statement:

"Uncertainties in the cloud ice retrieval may also be causing this discrepancy."

Your second point is difficult to believe as the temperatures are so high during the radiosonde launches. The only other possibility here than seeding would be that the cloud has previously been colder and thus formed ice that is then advected to the site. We can add this point:

"...or colder temperatures that allowed for primary ice nucleation upstream."

Line 354: "redundant" seems like the wrong word here.
→ The word redundant is simply to state that it does not make any sense to do further simulations, in regard to cloud ice sensitivity, when the temperature is too high for any of the parameterisations to be active. We will replace this with the synonym "unnecessary".

Line 425-437: This paragraph talks about impacts on frozen hydrometeors in a couple of ways. While there are some sensitivities, I would argue that these are rather inconsequential. First, in Fig. 10 the logarithmic Y axis means that much of the range and apparent changes are so small as to not even be detectable from the observational perspective. For the most extreme case of INPx1E6 the FWP only gets up to 1 g/m2! And the differences among the other cases are all very small and well below 1 g/m2. Thus, while indeed the model suggests that there is SOME ice present, it is inconsequential in most cases. Similarly, towards the end of the paragraph there is discussion of ice particle collisions. However, at the highest concentration there is 1 particle per 50 L, and for the 1E4 simulation it is 1 particle per 500L! Again, the number of collisions resulting from these low concentrations is vanishingly small and inconsequential for the cloud processes. It seems to me that this paragraph could be summarized by the statement that while some of these model changes have slight impacts, ice processes are not significant in any of them.
→The changes between the 1.6km simulation and 1E6 are not inconsequential as the

increase in FWP exceeds 3 orders of magnitude. Thus, we are confused as to which changes you deem inconsequental here.

We may agree with you that some values are very small in the discussion in the section surrounding Fig. 10. To this effect, we exclude Line 360-365 to remove some discussion surrounding very low values of snow/graupel production from changes in CCN concentration. However, we want to keep the discussion about the high INP scalings to explain the production of quite appreciable snow ($1gm^{-2}$) from the lower layer caused by the seeding from above (the collision discussion). We restructure the sentences to make it clear that the collision discussion is in regard to the seeding from above and not within the lower layer.

Line 457-458: As noted in the text, the key transition in the lower cloud occurs just after 12, but the process rates are given for 6-12 UTC. Thus, do the processes described in the figure actually represent the apparently abrupt shift that occurs just after 12?

→ No they do not, that is true. After 12, very rapid glaciation occurs which of course would increase the ice mass concentration greatly, however, this period is not the focus to us as, in effect, the mixed-phase cloud is destroyed. Furthermore, after 12UTC, in all simulations, the upper cloud dissipates before 13UTC which is why we limit the analysis to 12UTC.

Line 459-461: Perhaps I am missing a key detail here, but how do these SIP mechanisms impact the ice mass? These processes are ones in which individual ice crystals turn into multiple ice crystals, such that there would be an impact on ice number but not on mass (other than the conversion of liquid droplets to ice in some of the processes). Ice-ice collisions, for example, should have no mass change, just a number change. Thus, I'm confused by how these are put in terms of mass rates. I believe they should instead be number rates. If not, please provide a clear explanation for the reader.
→ It is correct that SIP does not directly increase the mass of the system as only higher numbers are generated through SIP. However, these fragments have an associated mass which is what is shown in Figure 11.

These fragments are then free to interact with their environment. We find that they impact the ice mass concentrations through higher vapour deposition rates, and subsequently, snow through higher aggregational rates. For clarity in terms of the SIP impacts, we will change the figure into rates of number concentrations (see below).

[Figure]

Fig. II: New Figure 11 with number concentrations of SIP rates. Please note the different ranges of the x-axis in panels c and f.

Line 468: Again, ice-ice breakup does not directly impact mass. The subsequent vapor growth of the new, higher number of ice crystals could impact mass growth, but that is no longer breakup but rather depositional growth.
→Please refer to the answer above.

Line 471-473: How much of a drop in temperature? This would be useful to know. Presumably this drop in temperature is due to advective tendencies because radiative cooling in the cloud layer should be minimal.
→We find an approximate 0.5°C reduction in temperature between 11-12UTC at cloud top. As the upper cloud is glaciating and dissipating at the same time, an increase in radiative cloud-top cooling could be expected in the lower layer.

Line 490: This is a nearly identical sentence to one that was included a couple of paragraphs prior.
→Yes thank you for noticing. However, this sentence is used to transition into the scalings of the breakup mechanism and thus we'd like to keep it for the justification of the scaling.

Line 491: Why is this not already "realistic"? What is the basis for thinking that breakup needs to be reduced?
→ Previous studies, as mentioned in Lines 186-191, believe this implementation of the breakup is not atmospherically relevant. For clarity, we may re-iterate this in the result section:

We add to Line 428:

"Previous modelling studies have deemed the contribution from the Takahashi et al. 1995 scheme too substantial and therefore introduce the concept of scaling this mechanism (Sotiropoulou et al. 2021, Georgakaki et al. 2022, Han et al. 2024). The scaled simulation (SIP scaled), whereby the breakup is scaled by the colliding particle diameter divided by the original particle diameter used in the experiments by Takahashi et al. 1995, nullifies the breakup contribution (Fig. 10 and contours in Fig.A1b)."

Line 495-500: I do not understand the SIP scaling. Is this simply an implementation of a different parameterization for collision breakup? If so, just be clear that it is a different parameterization that is not modified here but simply tested (and found to render collision breakup ineffective). My problem is that the word "scaling" makes it sound like there is some parameter that is being adjusted ("scaled") to modify the effectiveness of the collision process. If you believe the default collisional breakup is too aggressive (first that should be clarified and justified), it should certainly be possible to "scale" the collision process by adjusting one of multiple components of the collisional breakup parameterization. But I get the sense that the authors do not want to explore such details. In any case, the language used here is unclear. If I understand what is being said, perhaps it is most clear to just state that the so-called scaling adjustment to the collision parameterization effectively nullifies the process altogether.

→ The scaling is introduced in Line 191. The breakup parameterisation is scaled by the diameter of the colliding particles divided by the diameter used in the original experiment. Thus, we explore physical constraints to this parameterisation instead of tuning pre-factors that we cannot physically constrain. We further studied this scaling with a CCN constraint following comments from previous reviewers. The comment added above hopefully clarifies this point.

Line 534-536: Yes, I agree that the assimilation and/or advective tendencies for key parameters are likely problematic here. These may be why the liquid water is not well represented and not very responsive to the ice microphysics. It is worth noting this point explicitly here, unless there are other explanations for why the liquid is represented so poorly.

→ Yes, together with the saturation adjustment discussed above, this is likely a source of some errors. We will avoid further speculative sentences about the upstream processes as these were removed in a previous version of the paper in response to comments raised by reviewer #2 in the previous round of reviews.

We add: 'Improvements to the 1-moment scheme may further improve the cloud mass partitioning.'

Line 559-563: These statements are speculative and not supported by any evidence presented in this paper. i.e., I do not see any ICON-ART results that are shown to be more realistic for this type of case. Moreover, it has not been shown that better representation of updrafts, which might impact droplet activation, enhances the predictability of cloud properties. These statements should be removed. All that can really be said is that in future studies it might make sense to explore if ICON-ART and/or better resolved vertical motions can help to improve the representation of this type of cloud.

→These sentences were added as part of a previous reviewer's comments. We may, however, restructure them, highlighting their speculative nature.

We change the lines:

'Prognostic aerosols using a dynamic aerosol model such as ICON-ART (Aerosol and Reactive Trace gases module) *could potentially* improve the representation of the local CCN concentrations and provide a more realistic cloud droplet activation. Furthermore, accurate representation of updrafts *may* ensure a more realistic simulation of cloud droplet activation, which *could* lead to an enhanced predictability of cloud liquid and layering.

Line 585-586: This statement is not true. According to Fig. 10 the 1E6 simulation produces a reasonable amount of ice in both the upper and lower cloud layers, better than any of the SIP simulations. Thus, according to these simulations, SIP (at least the added SIP

parameterizations) are not required to reach observed levels of ice.

→ Yes, it does but that high level of INPs in this region is also quite unrealistic. We make this point a bit more clear;

"We find that a large scaling of this immersion freezing (by 1E6) captures observed levels of ice, however, the required abundance of INPs in the high-Arctic is unrealistic. We conclude that secondary ice production combined with increased primary ice production is required to reach observed levels of ice."

Line 606: The number of grid levels is increased, while the grid spacing is thus decreased.
→ Yes, thank you for noticing, we've changed it.

Appendix B: This appendix has no text, only figures. Does this conform to ACP style?

→We believe it does. Appendix A goes into detail about the high-resolution simulations, where some text is provided to give the full story while figures B1 and B2 serve as additional information to the main text. The moisture profiles in Fig. B1 are shown to remove doubts about vapour being the culprit of the high clouds during the 1st of September while B2 shows the process rates during the 3rd, these are sporadically mentioned to support the statements on the microphysical processes in the text.

Additional Note:
A typo was found where we state the IEF impact from the SIP. The correct values are now included in the revised MS. We have edited the IEF values from 10^5 to 10^6 and 10^4 to 10^5 which correspond to the values in Figure 11f.